# A Framework to Learn with Interpretation

**Jayneel Parekh** [1], **Pavlo Mozharovskyi** [1], **Florence d'Alché-Buc** [1]
[1] LTCI, Télécom Paris,
Institut Polytechnique de Paris, France
{jayneel.parekh,pavlo.mozharovskyi,florence.dalche}@telecom-paris.fr

## Abstract

To tackle interpretability in deep learning, we present a novel framework to jointly learn a predictive model and its associated interpretation model. The interpreter provides both local and global interpretability about the predictive model in terms of human-understandable high level attribute functions, with minimal loss of accuracy. This is achieved by a dedicated architecture and well chosen regularization penalties. We seek for a small-size dictionary of high level attribute functions that take as inputs the outputs of selected hidden layers and whose outputs feed a linear classifier. We impose strong conciseness on the activation of attributes with an entropy-based criterion while enforcing fidelity to both inputs and outputs of the predictive model. A detailed pipeline to visualize the learnt features is also developed. Moreover, besides generating interpretable models *by design*, our approach can be specialized to provide *post-hoc* interpretations for a pre-trained neural network. We validate our approach against several state-of-the-art methods on multiple datasets and show its efficacy on both kinds of tasks.

## 1   Introduction

Interpretability in machine learning systems [16, 37, 42] has recently attracted a large amount of attention. This is due to the increasing adoption of these tools in every area of automated decision-making, including critical domains such as law [25], healthcare [48] or defence. Besides robustness, fairness and safety, it is considered as an essential component to ensure trustworthiness in predictive models that exhibit a growing complexity. Explainability and interpretability are often used as synonyms in the literature, referring to the ability to provide human-understandable insights on the decision process. Throughout this paper, we opt for interpretability as in [15] and leave the term explainability for the ability to provide logical explanations or causal reasoning, both requiring more sophisticated frameworks [17, 19, 44]. To address the long-standing challenge of interpreting models such as deep neural networks [43, 10, 9], two main approaches have been developed in literature: *post-hoc* approaches and "*by design* methods".

Post-hoc approaches [7, 41, 38, 45] generally analyze a pre-trained system locally and attempt to interpret its decisions. "Interpretable by design" [3, 1] methods aim at integrating the interpretability objective into the learning process. They generally modify the structure of predictor function itself or add to the loss function regularizing penalties to enforce interpretability. Both approaches offer different types of advantages and drawbacks. Post-hoc approaches guarantee not affecting the performance of the pre-trained system but are however criticized for computational costs, robustness and faithfulness of interpretations [54, 28, 5]. Interpretable systems by-design on the other hand, although preferred for interpretability, face the challenge of not losing out on performance.

Here, we adopt another angle to learning interpretable models. As a starting point, we consider that prediction (computing $\hat{y}$ the model's output for a given input) and interpretation (giving a human-understandable description of properties of the input that lead to $\hat{y}$) are two distinct but strongly related tasks. On one hand, they do not involve the same criteria for the assessment of their

quality and might not be implemented using the same hypothesis space. On the other hand, we wish that an interpretable model relies on the components of a predictive model to remain faithful to it. These remarks yield to a novel generic task in machine learning called Supervised Learning with Interpretation (SLI). SLI is the problem of jointly learning a pair of dedicated models, a predictive model and an interpreter model, to provide both interpretability and prediction accuracy. In this work, we present FLINT (Framework to Learn With INTerpretation) as a solution to SLI when the model to interpret is a deep neural network classifier. The interpreter in FLINT implements the idea that a prediction to be understandable by a human should be linearly decomposed in terms of attribute functions that encode high-level concepts as other approaches [4, 18]. However, it enjoys two original key features. First the high-level attribute functions leverage the outputs of chosen hidden layers of the neural network. Second, together with expansion coefficients they are jointly learnt with the neural network to enable local and global interpretations. By local interpretation, we mean a subset of attribute functions whose simultaneous activation leads to the model's prediction, while by global interpretation, we refer to the description of each class in terms of a subset of attribute functions whose activation leads to the class prediction. Learning the pair of models involves the minimization of dedicated losses and penalty terms. In particular, local and global interpretability are enforced by imposing a limited number of attribute functions as well as conciseness and diversity among the activation of these attributes for a given input. Additionally we show that FLINT can be specialized to post-hoc interpretability if a pre-trained deep neural network is available.

**Key contributions:**

- We present FLINT devoted to Supervised Learning with Interpretation with an original interpreter network architecture based on some hidden layers of the network. The role of the interpreter is to provide local and global interpretability that we express using a novel notion of relevance of concepts.

- We propose a novel entropy and sparsity based criterion for promoting conciseness and diversity in the learnt attribute functions and develop a simple pipeline to visualize the encoded concepts based on previously proposed tools.

- We present extensive experiments on 4 image classification datasets, MNIST, FashionM-NIST, CIFAR10, QuickDraw, with a comparison with state-of-the-art approaches and a subjective evaluation study.

- Eventually, a specialization of FLINT to post-hoc interpretability is presented while corresponding numerical results are deferred to supplements.

## 2  Related Works

We emphasize here more on the methods relying upon a dictionary of high level attributes/concepts, a key feature of our framework. A synthetic view of this review is presented in the supplements to effectively view the connections and differences w.r.t wider literature regarding interpretability.

**Post-hoc interpretations.** Most works in literature focus on producing *a posteriori* interpretations for pre-trained models via input attribution. They often consider the model as a black-box [41, 38, 8, 31, 14] or in the case of deep neural networks, work with gradients to generate saliency maps for a given input [46, 47, 45, 40]. Very few post-hoc approaches rely on high level concepts or other means of interpretations [22]. Methods utilizing high level concepts come under the subclass of concept activation vector (CAV)-based approaches. TCAV [27] proposed to utilize human-annotated examples to represent concepts in terms of activations of a pre-trained neural network. The sensitivity of prediction to these concepts is estimated to offer an explanation. ACE [18] attempts to automate the human-annotation process by super-pixel segmentation and clustering these segments based on their perceptual similarity where each cluster represents a concept. ConceptSHAP [51] introduces the idea of "completeness" in ACE's framework. The CAV-based approaches already strongly differ from us in context of problem as they only consider post-hoc interpretations. TCAV generates candidate concepts using human supervision and not from the network itself. While ACE automates concept discovery, the concepts are less dynamic as by design they are associated to a single class and rely on being represented via spatially connected regions. Moreover, since ACE depends on using a CNN as perceptual similarity metric for image segments (regardless of aspect ratio, scale), it is limited in applicability (experimentally supported in supplement Sec. S.3).

**Interpretable neural networks by design.** Most works from this class learn a single model by either modifying the architecture [3], the loss functions [55, 13], or both [6, 36, 4, 12]. Methods like FRESH [24] and INVASE [53] perform selection over raw input tokens/features. The selected input features then are used by the final prediction model. GAME [35] shapes the learning problem as a co-operative game between predictor and interpreter. However, it learns a separate local interpreter for each sample rather than a single model. The above methods do not utilize high-level concepts for interpretation and offer local interpretations, with the exception of neural additive models [2], which are currently only suitable for tabular data.

Self Explaining Neural Networks (SENN) [4] presented a generalized linear model wherein coefficients are also modelled as a function of input. The linear structure is to emphasize interpretability. SENN imposes a gradient-based penalty to learn coefficients stably and other constraints to learn human understandable features. Unlike SENN, to avoid trade-off between accuracy and interpretability in FLINT, we allow the predictor to be an unrestrained neural network and jointly learn the interpreter. Interpretations are generated at a local and global level using a novel notion of relevance of attributes. Moreover, FLINT can be specialized for generating post-hoc interpretations of pre-trained networks.

**Known dictionary of concepts.** Some recent works have focused on different ways of utilizing a known dictionary of concepts for interpretability [26], by transforming the latent space to align with the concepts [13] or by adding user intervention as an additional feature to improve interactivity [29]. It should be noted that these methods are not comparable to FLINT or other interpretable networks by design as they assume availability of a ground truth dictionary of concepts for training.

## 3 Learning a classifier and its interpreter with FLINT

We introduce a novel generic task called *Supervised Learning with Interpretation* (SLI). Denoting $\mathcal{X}$ the input space, and $\mathcal{Y}$ the output space, we assume that the training set $\mathcal{S} = \{(x_i, y_i)_{i=1}^{N}\}$ is composed of $n$ independent realizations of a pair of random variables $(X, Y)$ defined over $\mathcal{X} \times \mathcal{Y}$. SLI refers to the idea that the **interpretation** task differs from the **prediction** task and must be taken over by a dedicated model that depends on the predictive model to be interpreted. Let us call $\mathcal{F}$ the space of predictive models from $\mathcal{X}$ to $\mathcal{Y}$. For a given model $f \in \mathcal{F}$, we denote $\mathcal{G}_f$ the family of models $g_f : \mathcal{X} \to \mathcal{Y}$, that depend on $f$ and are devoted to its interpretation. For sake of simplicity, an interpreter $g_f \in \mathcal{G}_f$ is denoted $g$, omitting the dependency on $f$. With these assumptions, the empirical loss of supervised learning is revisited to include explicitly an interpretability objective besides the prediction loss yielding to the following definition.

**Supervised Learning with Interpretation (SLI)**:

$$\textbf{Problem 1}: \arg \min_{f \in \mathcal{F}, g \in \mathcal{G}_f} \mathcal{L}_{pred}(f, \mathcal{S}) + \mathcal{L}_{int}(f, g, \mathcal{S}),$$

where $\mathcal{L}_{pred}(f, \mathcal{S})$ denotes a loss term related to prediction error and $\mathcal{L}_{int}(f, g, \mathcal{S})$ measures the ability of $g$ to provide interpretations of predictions by $f$.

The goal of this paper is to address Supervised Learning with Interpretation when the hypothesis space $\mathcal{F}$ is instantiated to deep neural networks and the task at hand is multi-class classification. We present a novel and general framework, called Framework to Learn with INTerpretation (FLINT) that relies on (i) a specific architecture for the interpreter model which leverages some hidden layers of the neural network network to be interpreted, (ii) notions of local and global interpretation and (iii) corresponding penalties in the loss function.

### 3.1 Design of FLINT

All along the paper, we take $\mathcal{X} = \mathbb{R}^d$ and $\mathcal{Y} = \{y \in \{0,1\}^C, \sum_{j=1}^{C} y^j = 1\}$, the set of $C$ one-hot encoding vectors of dimension $C$. We set $\mathcal{F}$ to the class of deep neural networks with $l$ hidden layers of respective dimension $d_1, \ldots, d_l$. Each element $f : \mathcal{X} \to \mathcal{Y}$ of $\mathcal{F}$ satisfies: $f = f_{l+1} \circ f_l \circ \ldots \circ f_1$ where $f_k : \mathbb{R}^{d_{k-1}} \to \mathbb{R}^{d_k}$, $d_0 = d, d_{l+1} = C, k = 1, \ldots, l+1$ is the function implemented by layer $k$. A network $f$ in $\mathcal{F}$ is completely identified by its generic parameter $\theta_f$. As for the interpreter model $g \in \mathcal{G}_f$, we propose the following original architecture which exploits the outputs of chosen hidden layers of $f$. Denote $\mathcal{I} = \{i_1, i_2, \ldots, i_T\} \subset \{1, \ldots, l\}$ the set of indices specifying the intermediate layers of network $f$ to be accessed and chosen for the representation of input. We define $D = \sum_{t=1}^{T} d_{i_t}$. Typically these layers are selected from the latter layers of the network $f$. The

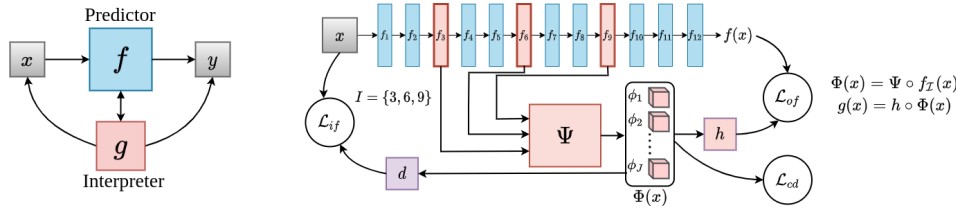

Figure 1: **(Left)** General view of FLINT. **(Right)** Instantiation of FLINT on a deep architecture.

concatenated vector of all intermediate outputs for an input sample $x$ is denoted as $f_{\mathcal{I}}(x) \in \mathbb{R}^D$. Given $f$ a network to be interpreted and a positive integer $J \in \mathbb{N}^*$, an **interpreter network** $g$ computes the composition of a dictionary of attribute functions $\Phi : \mathcal{X} \to \mathbb{R}^J$ and an interpretable function $h : \mathbb{R}^J \to \mathcal{Y}$.

$$\forall x \in \mathcal{X}, g(x) = h \circ \Phi(x), \tag{1}$$

In this work, we take: $h(\Phi(x)) := \text{softmax}(W^T \Phi(x))$ but other models like decision trees could be eligible. The **attribute dictionary** is composed of functions $\phi_j : \mathcal{X} \to \mathbb{R}^+, j = 1, \dots J$ whose non-negative images $\phi_j(x)$ can be interpreted as the activation of some high level attribute, i.e. a "concept" over $\mathcal{X}$. A key originality of the model lies in the fact that the attribute functions $\phi_j$ (referred to as attribute for simplicity) leverage the outputs of hidden layers of $f$ specified by $\mathcal{I}$:

$$\forall j \in \{1, \dots, J\}, \phi_j(x) = \psi_j \circ f_{\mathcal{I}}(x) \tag{2}$$

where each $\psi_j : \mathbb{R}^D \to \mathbb{R}^+$ operates on the accessed hidden layers. Here, the set of functions $\psi_j, j = 1, \dots J$ is defined to form a shallow network $\Psi$ (around 3 layers) whose output is $\Psi(f_{\mathcal{I}}(x)) = \Phi(x)$ (example architecture in Fig. 1). Interestingly, $\phi_j$ are defined over $\mathcal{X}$ and as a consequence can be interpreted in the input space which is the most meaningful for the user (see Sec. 4). For sake of simplicity, we denote $\Theta_g = (\theta_\Psi, \theta_h)$ the specific parameters of this model, while the parameters devoted to the computation of $f_{\mathcal{I}}(x)$ are shared with $f$.

## 3.2 Interpretation in FLINT

The interpreter being defined, we need to specify its expected role and corresponding interpretability objective. In FLINT, interpretation is seen as an additional task besides prediction. We are interested by two kinds of interpretation, one at the global level that helps to understand which attribute functions are useful to predict a class and the other at the local level, that indicates which attribute functions are involved in prediction of a specific sample. As a preamble, note that, to interpret a local prediction $f(x)$, we require that the interpreter output $g(x)$ matches $f(x)$. When the two models disagree, we provide a way to analyze the conflictual data and possibly raise an issue about the confidence on the prediction $f(x)$ (see Supplementary Sec. S.2). To define local and global interpretation, we rely on the notion of relevance of an attribute.

Given an interpreter with parameter $\Theta_g = (\theta_\Psi, \theta_h)$ and some input $x$, the **relevance score** of an attribute $\phi_j$ is defined regarding the prediction $g(x) = f(x) = \hat{y}$. Denoting $\hat{y} \in \mathcal{Y}$ the index of the predicted class and $w_{j,\hat{y}} \in W$ the coefficient associated to this class, the contribution of attribute $\phi_j$ to unnormalized score of class $\hat{y}$ is $\alpha_{j,\hat{y},x} = \phi_j(x).w_{j,\hat{y}}$. The relevance score is computed by normalizing contribution $\alpha$ as $r_{j,x} = \frac{\alpha_{j,\hat{y},x}}{\max_i |\alpha_{i,\hat{y},x}|}$. An attribute $\phi_j$ is considered as relevant for a local prediction if it is both activated and effectively used in the linear (logistic) model. The notion of relevance of an attribute for a sample is extended to its "overall" importance in the prediction of any class $c$. This can be done by simply averaging relevance scores from local interpretations over a random subset or whole of the training set $\mathcal{S}$, where predicted class is $c$. Thus, we have: $r_{j,c} = \frac{1}{|\mathcal{S}_c|} \sum_{x \in \mathcal{S}_c} r_{j,x}, \mathcal{S}_c = \{x \in \mathcal{S} | \hat{y} = c\}$. Now, we can introduce the notions of local and global interpretations that the interpreter will provide.

**Definition 1 (Global and Local Interpretation)** *For a prediction network $f$, the* **global interpretation** *$G(g, f)$ provided by an interpreter $g$, is the set of class-attribute pairs $(c, \phi_j)$ such that their global relevance $r_{j,c}$ is greater than some threshold $1/\tau, \tau > 1$. A* **local interpretation** *for a sample $x$ provided by an interpreter $g$ of $f$ denoted $L(x, g, f)$ is the set of attribute functions $\phi_j$ with local relevance score $r_{j,x}$ greater than some threshold $1/\tau, \tau > 1$.*

It is important to note that these definitions do not prejudge the quality of local and global interpretations. Next, we convert desirable properties of the interpreter into specific loss functions.

### 3.3 Learning by imposing interpretability properties

Although converting desirable interpretability properties into losses is shared by several by-design approaches [11, 38], there is no consensus on these properties. We propose below a minimal set of penalties which are suitable for the proposed architecture and sufficient to provide relevant interpretations.

**Fidelity to Output.** The output of the interpreter $g(x)$ should be "close" to $f(x)$ for any $x$. This can be imposed through a cross-entropy loss:

$$\mathcal{L}_{of}(f, g, \mathcal{S}) = - \sum_{x \in \mathcal{S}} h(\Psi(f_{\mathcal{I}}(x)))^T \log(f(x))$$

**Conciseness and Diversity of Interpretations.** For any given sample $x$, we wish to get a *small* number of attributes in its associated local interpretation. This property of *conciseness* should make the interpretation easier to understand due to fewer attributes to be analyzed and promote the "high-level" character in the encoded concepts. However, to encourage better use of available attributes we also expect activation of multiple attributes across many randomly selected samples. We refer to this property as *diversity*. This is also important to avoid the case of attribute functions being learnt as class exclusive (for eg. reshuffled version of class logits). To enforce these conditions we utilize notion of entropy defined for real vectors proposed by Jain et al [23] to solve problem of efficient image search. For a real-valued vector $v$, the entropy is defined as $\mathcal{E}(v) = - \sum_i p_i \log(p_i)$, $p_i = \exp(v_i)/(\sum_i \exp(v_i))$.

Conciseness is promoted by minimizing $\mathcal{E}(\Psi(f_{\mathcal{I}}(x)))$ and diversity is promoted by maximizing entropy of average $\Psi(f_{\mathcal{I}}(x))$ over a mini-batch. Note that this can be seen as encouraging the interpreter to find a sparse and diverse coding of $f_{\mathcal{I}}(x)$ using the function $\Psi$. Since entropy-based losses have inherent normalization, they do not constrain the magnitude of the attribute activation. This often leads to poor optimization. Thus, we also minimize the $\ell_1$ norm $\|\Psi(f_{\mathcal{I}}(x))\|_1$ (with hyperparameter $\eta$) to avoid it. Note that $\ell_1$-regularization is a common tool to encourage sparsity and thus conciseness, however we show in the experiments that entropy provides a more effective way.

$$\mathcal{L}_{cd}(f, g, \mathcal{S}) = -\mathcal{E}(\bar{\Phi}_{\mathcal{S}}) + \sum_{x \in \mathcal{S}} \mathcal{E}(\Psi(f_{\mathcal{I}}(x))) + \sum_{x \in \mathcal{S}} \eta \|\Psi(f_{\mathcal{I}}(x))\|_1 \quad \text{with} \quad \bar{\Phi}_{\mathcal{S}} = \frac{1}{|\mathcal{S}|} \sum_{x \in \mathcal{S}} \Psi(f_{\mathcal{I}}(x))$$

**Fidelity to Input.** To encourage encoding high-level patterns related to input in $\Phi(x)$, we use a decoder network $d : \mathbb{R}^J \to \mathcal{X}$ that takes as input the dictionary of attributes $\Psi(f_{\mathcal{I}}(x))$ and reconstructs $x$. A similar penalty has previously been applied by [4].

$$\mathcal{L}_{if}(f, g, d, \mathcal{S}) = \sum_{x \in \mathcal{S}} (d(\Psi(f_{\mathcal{I}}(x))) - x)^2$$

Note that one can modify $\mathcal{L}_{if}$ with other reconstruction losses as well (such as $\ell_1$-reconstruction).

Given the proposed loss terms, the loss for interpretability writes as follows:

$$\mathcal{L}_{int}(f, g, d, \mathcal{S}) = \beta \mathcal{L}_{of}(f, g, \mathcal{S}) + \gamma \mathcal{L}_{if}(f, g, d, \mathcal{S}) + \delta \mathcal{L}_{cd}(f, g, \mathcal{S})$$

where $\beta, \gamma, \delta$ are non-negative hyperparameters. The total loss to be minimized $\mathcal{L} = \mathcal{L}_{pred} + \mathcal{L}_{int}$, where the prediction loss, $\mathcal{L}_{pred}$, is the well-known cross-entropy loss.

Let us denote $\Theta = (\theta_f, \theta_d, \theta_\Psi, \theta_h)$ the parameters of these networks. Learning the models $f$, $\Psi$, $h$ and $d$ boils down to learning $\Theta$. In practice, introducing all the losses at once often leads to very poor optimization. Thus, we follow the procedure described in Alg. 1. We train the networks with $\mathcal{L}_{pred}, \mathcal{L}_{if}$ for the first two epochs and gain a reasonable level of accuracy. From the third epoch we introduce $\mathcal{L}_{of}$ and from the fourth epoch we introduce $\mathcal{L}_{cd}$ loss.

## 4 Understanding encoded concepts in FLINT

Once the predictor and interpreter are jointly learnt, interpretation can be given at the global and local levels as in Def. 1. A key component to grasp the interpretations is to understand the concept

---

**Algorithm 1** Learning algorithm for FLINT

---

1: **Input:** $\mathcal{S}$ & parameters $\Theta = (\theta_f, \theta_d, \theta_\Psi, \theta_h)$ & hyperparameters: $\beta_0, \gamma_0, \delta_0, \eta_0$ & number of batches $B$ & number of training epochs $N_{epoch}$.
2: Random initialization of parameter $\Theta_0$
3: $\Theta_1 \leftarrow$ Train $(\mathcal{S}, \Theta_0, \beta = 0, \gamma_0, \delta = 0, \eta = 0, B, 2)$ {% Trains 2 epochs with $\mathcal{L}_{pred}, \mathcal{L}_{if}$}
4: $\Theta_2 \leftarrow$ Train $(\mathcal{S}, \Theta_1, \beta = \beta_0, \gamma_0, \delta = 0, \eta = 0, B, 1)$ {% Trains 1 epoch with $\mathcal{L}_{pred}, \mathcal{L}_{if}, \mathcal{L}_{of}$}
5: $\hat{\Theta} \leftarrow$ Train $(\mathcal{S}, \Theta_2, \beta_0, \gamma_0, \delta_0, \eta_0, B, N_{epoch} - 3)$ {% Trains with all losses}
6: **Output:** $\hat{\Theta} = (\hat{\theta}_f, \hat{\theta}_d, \hat{\theta}_\Psi, \hat{\theta}_h)$

---

encoded by each individual attribute function $\phi_j$, previously defined in Eq. 2. In this work, we focus on image classification and propose to represent an encoded concept as a set of visual patterns in the **input space** which highly activate $\phi_j$. We present a pipeline to generate visualizations for global and local interpretation by adapting various previously proposed tools [4, 39].

---

**Algorithm 2** Visualization of global interpretation

---

1: **Input:** (class,attribute):$(c, \phi_j)$ & subset size:$l$ & training set:$\mathcal{S}_n$ & AM+PI params:$(\lambda_\phi, \lambda_{tv}, \lambda_{bo})$
2: $\mathcal{S}_c = \{x | (x, c) \in \mathcal{S}_n\}$
3: $\text{MAS}(c, \phi_j, l) \leftarrow \arg\max_{\mathcal{M} \subset \mathcal{S}_c, |\mathcal{M}| = l} \sum_{x_i \in \mathcal{M}} \phi_j(x)$
4: FOR $x_k \in \text{MAS}(c, \phi_j, l)$
5: $x_{vis}^k \leftarrow \text{AM+PI}(x_k, \lambda_\phi, \lambda_{tv}, \lambda_{bo})$
6: ENDFOR
7: **Output:**$\{x_{vis}^1, \dots, x_{vis}^l\}, \text{MAS}(c, \phi_j, l)$

---

**Visualization of global interpretation.** Given any class-attribute pair $(c, \phi_j)$ in the global interpretation $G(g, f)$, we first select a small subset of training samples from class $c$ that maximally activate $\phi_j$. This set of samples is referred to as maximum activating samples and denoted $\text{MAS}(c, \phi_j, l)$ where $l$ is the size of the subset (chosen as 3 in the experiments). Although, MAS reveal some information about the encoded concept, it might not be apparent what aspect of these samples causes activation of $\phi_j$. We thus propose further analyzing each element in MAS through tools that enhance the detected concept. This results in a much better understanding. The primary tool we employ is a modified version of activation maximization [39], which we refer to as *activation maximization with partial initialization* (AM+PI).

Given a maximum activating sample $x' \in \text{MAS}(c, \phi_j, l)$, the key idea behind AM+PI is to synthesize appropriate input via optimization, that maximally activates $\phi_j$. We thus optimize a common activation maximization objective [39]: $\arg\max_x \lambda_\phi \phi_j(x) - \lambda_{tv}\text{TV}(x) - \lambda_{bo}\text{Bo}(x)$ , where $\text{TV}(.), \text{Bo}(.)$ are regularization terms. However, we initialize the procedure by low-intensity version of sample $x'$. This makes the optimization easier with the detected concept weakly present in the input. This also allows the optimization to "fill" the input to enhance the encoded concept. As an output, we obtain a map adapted to $x'$, that strongly activates $\phi_j$. Complete details of the AM+PI procedure are given in supplementary (Sec. S.2). Visualization of a class-attribute pair is summarized in Alg. 2. Alternative useful tools are discussed in the supplementary (Sec. S.2).

**Local analysis.** Given any test sample $x_0$, one can determine its local interpretation $L(x_0, f, g)$, the set of relevant attribute functions accordingly to Def. 1. To visualize a relevant attribute $\phi_j \in L(x_0, f, g)$, we can repeat the AM+PI procedure with initialization using low-intensity version of $x_0$ to enhance concept detected by $\phi_j$ in $x_0$. Note that the understanding built about any attribute function $\phi_j$ via global analysis, although not essential, can still be helpful to understand the generated AM+PI maps during local analysis, as these maps are generally similar.

## 5 Numerical Experiments for FLINT

**Datasets and Networks.** We consider 4 datasets for experiments, MNIST [34], FashionMNIST [50], CIFAR-10 [30], and a subset of QuickDraw dataset [20]. Additional results on CIFAR100 [30] (large number of classes) and Caltech-UCSD Birds-200-2011 [49] (large-scale images and large number of

| | Accuracy (in %) | | | | | Fidelity (in %) | | |
|---|---|---|---|---|---|---|---|---|
| | BASE-$f$ | SENN | PrototypeDNN | FLINT-$f$ | FLINT-$g$ | LIME | VIBI | FLINT-$g$ |
| MNIST | 98.9±0.1 | 98.4±0.1 | **99.2** | 98.9±0.2 | 98.3±0.2 | 95.6±0.4 | 96.6±0.7 | **98.7±0.1** |
| FashionMNIST | 90.4±0.1 | 84.2±0.3 | 90.0 | **90.5±0.2** | 86.8±0.4 | 67.3±1.3 | 88.4±0.3 | **91.5±0.1** |
| CIFAR10 | 84.7±0.3 | 77.8±0.7 | – | **84.5±0.2** | 84.0±0.4 | 31.5±0.9 | 65.5±0.3 | **93.2±0.2** |
| QuickDraw | 85.3±0.2 | 85.5±0.4 | – | **85.7±0.3** | 85.4±0.1 | 76.3±0.1 | 78.6±0.4 | **90.8±0.4** |

Table 1: Results for accuracy (in %) and fidelity to FLINT-$f$ on different datasets. BASE-$f$ is system trained with just accuracy loss. FLINT-$f$, FLINT-$g$ denote the predictor and interpreter trained in our framework. Mean and standard deviation of 4 runs for each system are reported

classes) are covered in supplementary (Sec. S.2.2). Our experiments include 2 kinds of architectures for predictor $f$: (i) LeNet-based [33] network for MNIST, FashionMNIST, and (ii) ResNet18-based [21] network for QuickDraw, CIFAR. We select one intermediate layer for LeNet based network and two for ResNet based networks, from the last few convolutional layers as they are expected to capture higher-level features. We set the number of attributes $J = 25$ for MNIST, FashionMNIST, $J = 24$ QuickDraw and $J = 36$ for CIFAR. Further details about the QuickDraw subset, precise architecture, ablation studies about choice of hyperparameters (hidden layers, size of attribute dictionary, loss scheduling) and optimization details are available in supplementary (Sec. S.2).

## 5.1 Quantitative evaluation of FLINT

We evaluate and compare our model with other state-of-the-art systems regarding accuracy and interpretability. The evaluation metrics for interpretability [15] are defined to measure the effectiveness of the losses proposed in Sec. 3.3. Our primary method for comparison, wherever applicable, is SENN, as it is an interpretable network by design with same units for interpretation as FLINT. Other baselines include PrototypeDNN [36] for predictive performance, LIME [41] and VIBI [8] for fidelity of interpretations. Implementation of our method is available on Github [1]. Details for implementation of baselines are in supplementary (Sec. S.2).

**Predictive performance of FLINT.** There are two goals to validate related to predictor trained with FLINT (denoted FLINT-$f$), (i) Jointly training $f$ with $g$ and backpropagating loss term $\mathcal{L}_{int}$ does not negatively impact performance, and (ii) The achieved performance is comparable with other similar interpretable by-design models. For the former we compare the accuracy of FLINT-$f$ with same predictor architecture trained just with $\mathcal{L}_{pred}$ (denoted by BASE-$f$). For the latter goal we compare accuracy of FLINT-$f$ with accuracy of SENN and another interpretable network by design PrototypeDNN [36] that does not use input attribution for interpretations. Note that PrototypeDNN requires non-trivial changes to the model for running on more complex datasets, CIFAR10 and QuickDraw. To avoid any unfair comparison we skip these results. The accuracies are reported in Tab. 1. They indicate that training $f$ within FLINT does not result in any significant accuracy loss on any dataset. Also, FLINT is competitive with other interpretable by-design models.

**Fidelity of Interpreter.** The fraction of samples where prediction of a model and its interpreter agree, i.e predict the same class, is referred to as *fidelity*. It is a commonly used metric to measure how well an interpreter approximates a model [8, 32]. Note that, typically, for interpretable by design models, fidelity cannot be measured as they only consider a single model. However, to validate that the interpreter trained with FLINT (denoted as FLINT-$g$) achieves a reasonable level of agreement with FLINT-$f$, we benchmark its fidelity against a state-of-the-art black-box explainer VIBI [8] and a traditional method LIME [41]. The results for this are provided in Tab. 1 (last three columns). FLINT-$g$ consistently achieves higher fidelity. Even though it is not a fair comparison as other systems are black-box explainers and FLINT-$g$ accesses intermediate layers, they clearly show that FLINT-$g$ demonstrates high fidelity to FLINT-$f$.

**Conciseness of interpretations.** We evaluate conciseness by measuring the average number of *important* attributes in generated interpretations. For a given sample $x$, it can be computed as number of attributes $\phi_j$ with $r_{j,x}$ greater than a threshold $1/\tau, \tau > 1$, i.e. $\text{CNS}_{g,x} = |\{j : |r_{j,x}| > 1/\tau\}|$. For different thresholds $1/\tau$, we compute the mean of $\text{CNS}_{g,x}$ over test data to estimate conciseness of $g$, $\text{CNS}_g$. Lower conciseness indicates need to analyze a lower number of attributes on an

---
[1] `https://github.com/jayneelparekh/FLINT`

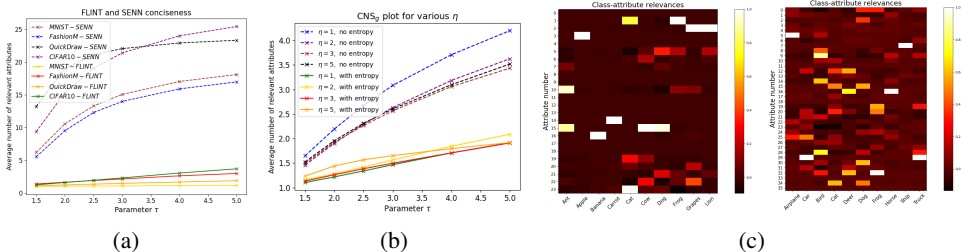

|       |       |       |
| :---: | :---: | :---: |
| (a) | (b) | (c) |

Figure 2: (a) Conciseness comparison of FLINT and SENN. (b) Effect of entropy losses on conciseness of ResNet for QuickDraw for various $\ell_1$-regularization levels. (c) Global class-attribute relevances $r_{j,c}$ for QuickDraw (Left) and CIFAR10 (Right). 24 class-attribute pairs for QuickDraw and 32 pairs for CIFAR10 have relevance $r_{j,c} > 0.2$.

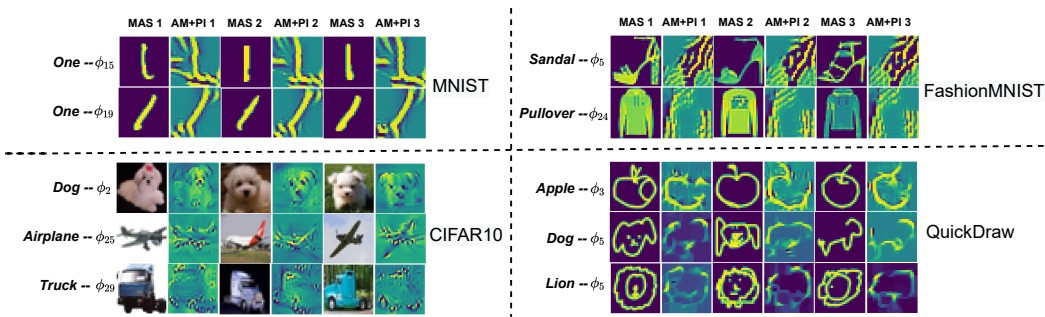

Figure 3: Example class-attribute pair analysis on all datasets, with global relevance $r_{j,c} > 0.2$. Each row contains 3 MAS with corresponding AM+PI outputs

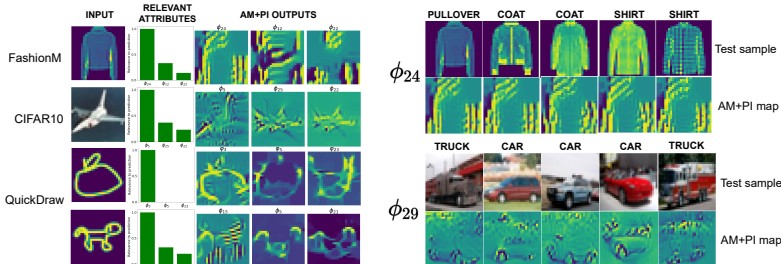

Figure 4: **(Left)** Local interpretations for test samples. Top 3 attributes with corresponding AM+PI output are shown. True labels for inputs are: Pullover, Airplane, Apple, Dog. **(Right)** Examples of attribute functions detecting same part across various test samples. For each sample, their relevance is greater than 0.8. True labels of samples indicated above them.

average. SENN is the only other system for which this curve can be computed. We thus compare the conciseness of SENN with FLINT on all four datasets. Fig. 2a depicts the same. It can be easily observed that FLINT produces lot more concise interpretations compared to SENN. Moreover, SENN even ends up with majority of concepts being considered relevant for lower thresholds (higher $\tau$).

**Entropy vs $\ell_1$ regularization.** We validate the effectiveness of entropy losses by computing conciseness curve at various levels of $\ell_1$ regularization strength, with and without entropy, for ResNet with QuickDraw. This is reported in Fig. 2b. The figure confirms that using the entropy-based loss is more effective in inducing conciseness of explanations compared to using just $\ell_1$-regularization, with the difference being close to use of 1 attribute less when entropy losses are employed.

**Importance of attributes.** Additional experiments evaluating meaningfulness of attributes by shuffling them and observing the effect (for FLINT and SENN) are given in supplementary (Sec. S.2).

## 5.2 Qualitative analysis

**Global interpretation.** Fig. 2c depicts the generated global relevances $r_{j,c}$ for all class-attribute pairs on QuickDraw and CIFAR. Each class-attribute pair with 'high' relevance needs to be analyzed as part of global analysis. Some example class-attribute pairs, with high relevance, are visualized in Fig. 3. For each pair we select MAS of size 3 and also show their AM+PI outputs. As mentioned before, simply analyzing MAS reveals useful information about the encoded concept. For instance, based on MAS, $\phi_{15}$, $\phi_{19}$ on MNIST, relevant for class 'One', clearly seem to activate for vertical and diagonal strokes respectively. However, AM+PI outputs give deeper insights about the concept by revealing more clearly what parts of input activate an attribute function. For eg., while MAS indicate that $\phi_5$ on FashionMNIST activates for heels (one type of 'Sandal'), $\phi_2$ on CIFAR10 activates for white dogs, it is not clear what part the attribute focuses on. AM+PI outputs indicate that $\phi_2$ focuses on the area around eyes and nose (the most enhanced regions), $\phi_5$ primarily detects a thin diagonal stroke of the heel surrounded by empty space. AM+PI outputs generally become even more important for attributes relevant for multiple classes. One such example is the function $\phi_5$ on QuickDraw, relevant for both 'Dog' and 'Lion'. It activates for very similar set of strokes for all samples, as indicated by AM+PI maps. For 'Dog' this corresponds to ears and mouth and for 'Lion' it corresponds to the mane. Other such attribute functions in the figure include $\phi_{24}$ on FashionMNIST, relevant for 'Pullover', 'Coat' and 'Shirt' which detects long sleeves and $\phi_{29}$ on CIFAR10, relevant for 'Trucks', 'Cars' and primarily detects wheels and parts of upper body. Further visualizations including those of other relevant classes for $\phi_{24}$, $\phi_{29}$ and global relevances are available in supplementary (Sec. S.2).

**Local interpretation.** Fig. 4 (left) displays the local interpretation visualizations for test samples. $f$ and $g$ both predict the true class in all the cases. We show the top 3 relevant attributes to the prediction with their relevances and their corresponding AM+PI outputs. Based on the AM+PI outputs it can be observed that the attribute functions generally activate for patterns corresponding to the same concept as inferred during global analysis. This can be easily seen for attribute functions present in both Fig. 3, 4 (left). This is further illustrated by Fig. 4 (right) where we illustrate AM+PI outputs for two attributes from Fig. 3. These functions are relevant for more than one class and detect the same concept across various test samples, namely long sleeves for $\phi_{24}$ and primarily wheels for $\phi_{29}$.

## 5.3 Subjective evaluation

We conducted a *survey based subjective evaluation* with QuickDraw dataset for FLINT with 20 respondents. We selected 10 attributes, covering 17 class-attribute pairs from the QuickDraw dataset. For each attribute we present the respondent with our visualizations (3 MAS and AM+PI outputs) for each of its relevant classes along with a textual description. We ask them if the description meaningfully associates to patterns in the AM+PI outputs. They indicate level of agreement with choices: Strongly Agree (SA), Agree (A), Disagree (D), Strongly Disagree (SD), Don't Know (DK). Descriptions were manually generated by our understanding of encoded concept for each attribute. 40% incorrect descriptions were carefully included to ensure informed responses. These were forcefully related to the classes shown to make them harder to identify. **Results** – for correct descriptions: 77.5% – SA/A, 10.0% – DK, 12.5% – D/SD. For incorrect descriptions: 83.7% – D/SD, 7.5% – DK, 8.8% – SA/A. These results clearly indicate that concepts encoded in FLINT's learnt attributes are understandable to humans. Survey details are given in supplementary (Sec. S.2).

## 6 Specialization of FLINT to post-hoc interpretability

While interpretability by design is the primary goal of FLINT, it can be specialized to provide a *post-hoc* interpretation when a classifier $\hat{f}$ is already available. The **Post-hoc interpretation learning** (see for instance [41]) comes as a special case of SLI and is defined as follows. Given a classifier $\hat{f} \in \mathcal{F}$ and a training set $\mathcal{S}$, the goal is to build an interpreter of $\hat{f}$ by solving:

$$\textbf{Problem 2}: \arg\min_{g \in \mathcal{G}_{\hat{f}}} \mathcal{L}_{int}(\hat{f}, g, \mathcal{S}).$$

With FLINT, we have $g(x) = h \circ \Phi(x)$ and $\Phi(x) = \Psi \circ \hat{f}_{\mathcal{I}}(x)$ for a given set of accessible hidden layers $\mathcal{I}$ and a attribute dictionary size $J$. Learning can be performed by specializing Alg. 1 with slight modification of replacing $\Theta$ as $\Theta = (\theta_\Psi, \theta_h, \theta_d)$ while $\theta_{\hat{f}}$ is fixed and eliminating $\mathcal{L}_{pred}$ from training loss $\mathcal{L}$.

**Experimental results for post-hoc FLINT:** We validate this ability of our framework by interpreting fixed models trained only for accuracy, i.e, BASE-$f$ models from section 5.1. Even after not tuning the internal layers of $f$, the system is still able to generate high-fidelity and meaningful interpretations. Fidelity comparisons against VIBI, class-attribute pair visualizations and experimental details are available in supplementary (Sec. S.3).

# 7 Discussion and Perspectives

FLINT is a novel framework for learning a predictor network and its interpreter network with dedicated losses. It provides local and global interpretations in terms of high-level learnt attributes/concepts by relying on (some) hidden layers of the prediction network. This however leaves some under-explored questions about *faithfulness* of interpretations to the predictor. Defining faithfulness of an interpretation regarding a decision process [4] has not yet reached a consensus particularly in the case of post-hoc interpretability or when the two models, predictor and interpreter, differ [52]. Even though generating interpretations based on hidden layers of predictor ensures high level of faithfulness of the interpreter to the predictor, a complete faithfulness cannot be guaranteed since predictor and interpreter differ in their last part. However if ensuring faithfulness by design is regarded as the primary objective, nothing stops the use of interpreter FLINT-$g$ as the final decision-making network. In this case, there is only one network and the so-called prediction network has only played the useful role of providing relevant hidden layers.

Retaining only the interpreter model additionally provides a compression of the predictor and can be relevant when frugality is at stake. Further works will investigate this direction and the enforcement of additional constraints on attribute functions to encourage invariance under various transformations. Eventually FLINT can be extended to other tasks or modalities other than images in particular by adapting the design of attributes and the pipeline to understand them.

## Acknowledgments and Disclosure of Funding

This work has been funded by the research chair Data Science & Artificial Intelligence for Digitalized Industry and Services of Télécom Paris. This work also benefited from the support of French National Research Agency (ANR) under reference ANR-20-CE23-0028 (LIMPID project). The authors would like to thank Sanjeel Parekh for fruitful discussions and anonymous reviewers for their valuable comments and suggestions.

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
