## S.1 Overview of related works

To recap the properties of the methods exposed in Sec. 2 (main paper), we provide in Tab. 1 a synthetic view of the major properties of interpretable methods along three aspects. *Type* denotes if the method implements *post-hoc* interpretations for a trained model or interpretable models *by-design*). *Scope* reflects the ability of the approach to provide interpretation of decisions for individual samples (*Local*) or to understand the model as a whole (*Global*). *Means* denotes the units in which the interpretations are generated. Categories include raw input features, a simplified representation of input, logical rules, prototypes, high-level concepts.

| System | Means | Type | Scope |
|---|---|---|---|
| LIME, SHAP | Simplified input | Post-hoc | Local+Global |
| Gradient based | Raw input | Post-hoc | Local |
| VIBI, L2X | Raw input | Post-hoc | Local |
| Anchors | Logical rules | Post-hoc | Local |
| ICNN | Raw input | By-design | Local |
| INVASE | Raw input | By-design | Local |
| CEN, GAME | Simplified input | By-design | Local |
| PrototypeDNN | Prototypes | By-design | Local |
| CAV-based | Concepts (External) | Post-hoc | Local+Global |
| SENN | Concepts (Learnt) | By-design | Local |
| FLINT | Concepts (Learnt) | Both | Local+Global |

Table 1: Various interpretability systems and their properties.

- LIME, SHAP: Local Interpretable Model-agnostic Explanations [17], SHapley Additive exPlanations [14].
- VIBI, L2X: Variational Information Bottleneck for Interpretation [3], Learning to Explain [4].
- ICNN: Interpretable CNN [23].
- INVASE: Instance-Wise Variable Selection using Neural Networks [22].
- CEN, GAME: Contextual Explanation Networks [1], Game-theoretic transparency[12].
- PrototypeDNN: [13].
- Anchors: [18].
- CAV-based: Testing with Concept Activation Vectors (TCAV) [8], Towards Automatic Concept-based Explanations (ACE) [5], ConceptSHAP [21].
- SENN: Self Explaining Neural Networks [2].

## S.2 Interpretability by design: Additional information and experiments

We cover all the implementation details in Sec. S.2.1, including network architectures, choice of hyperparameters, optimization procedures, resource consumption. Experiments on CIFAR100 and

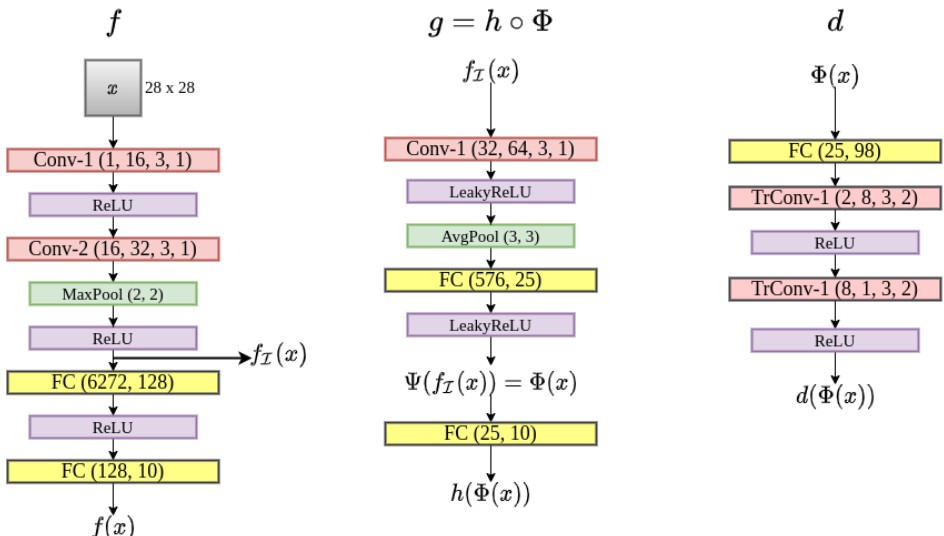

Figure 1: Architecture of networks based on LeNet [11]. Conv (a, b, c, d) and TrConv (a, b, c, d) denote a convolutional, transposed convolutional layer respectively with number of input maps a, number of output maps b, kernel size c × c and stride size d. FC(a, b) denotes a fully-connected layer with number of input neurons a and output neurons b. MaxPool(a, a) denotes window size a × a for the max operation. AvgPool(a, a) denotes the output shape a × a for each input map

CUB-200 are detailed in Sec. S.2.2. Additional analysis including ablation studies and visualizations for attributes are available in Sec. S.2.3. We also present other useful tools for analysis in Sec. S.2.4. Baseline implementations are discussed in Sec. S.2.5. Details about the subjective evaluation, including the form link are available in Sec. S.2.6. Note that the experiments with ACE are deferred to Sec. S.3.3.

### S.2.1   Implementation details

#### S.2.1.1   Network architectures

**Predictor**    Fig. 1 and 2 depict the architectures used for experiments with predictor architecture based on LeNet [11] (on MNIST, Fashion-MNIST) and ResNet18 (on CIFAR10, QuickDraw) [7] respectively.

**Interpreter**    The architecture of interpreter $g = h \circ \Phi$ and decoder $d$ for MNIST, FashionMNIST are shown in Fig. 1. Corresponding architectures for QuickDraw are in Fig. 2. For CIFAR-10, the interpreter architecture is almost exactly the same as QuickDraw, with only difference being output layer for $\Phi(x)$, which contains 36 attributes instead of 24. The decoder $d$ also contains corresponding changes to input and output FC layers, with 36 dimensional input in first FC layer and 3072 dimensional output in last FC layer.

The choice of selection of intermediate layers is an interesting part of designing the interpreter. In case of LeNet, we select the output of final convolutional layer. For ResNet, while we tend to select the intermediate layers from the latter convolutional layers, we do not select the last convolutional block (CBlock 8) output. This is mainly because empirically, when selecting the output of CBlock 8, the attributes were trivially learnt, with only one attribute activating for any sample and attributes exclusively activating for a single class. The hyperparameters are much harder to tune to avoid this scenario. Thus we selected two outputs from CBlock 6, CBlock 7 as intermediate layers. The layers in the interpreter itself were chosen fairly straightforwardly with 1-2 conv layers followed by a pooling and fully-connected layer.

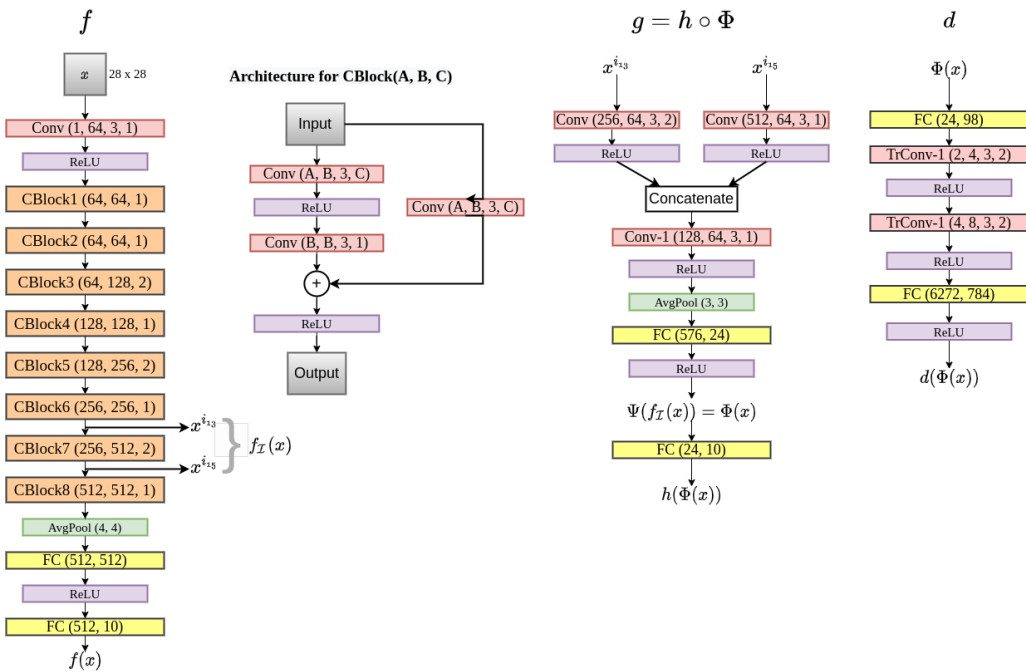

Figure 2: Architecture of networks for experiments on QuickDraw with network based on ResNet [7]. Conv (a, b, c, d) and TrConv (a, b, c, d) denote a convolutional, transposed convolutional layer respectively with number of input maps a, number of output maps b, kernel size c × c and stride size d. FC(a, b) denotes a fully-connected layer with number of input neurons a and output neurons b. AvgPool(a, a) denotes the output shape a × a for each input map. Notation for CBlock is explained in the figure.

### S.2.1.2 QuickDraw subset and pre-processing

**QuickDraw.** We created a subset of QuickDraw from the original dataset [6], by selecting 10000 random images from each of 10 classes: 'Ant', 'Apple', 'Banana', 'Carrot', 'Cat', 'Cow', 'Dog', 'Frog', 'Grapes', 'Lion'. We randomly divide each class into 8000 training and 2000 test images.

**Input pre-processing.** For MNIST, FashionMNIST and QuickDraw, we use the default images with pixel values in range $[0, 1]$. No data augmentation is performed. For CIFAR-10 we apply the most common mean and standard deviation normalization. The training data is generated by randomly cropping a $32 \times 32 \times 3$ image after padding the original images by zeros (size of padding is 2).

### S.2.1.3 Hyperparameter settings

| Variable | MNIST | FashionM | CIFAR10 | QuickDraw |
|---|---|---|---|---|
| $N_{epoch}$ – Number of training epochs | 12 | 12 | 25 | 12 |
| $\beta$ – Weight for $\mathcal{L}_{of}$ | 0.5 | 0.5 | 0.6 | 0.1 |
| $\gamma$ – Weight for $\mathcal{L}_{if}$ | 0.8 | 0.8 | 2.0 | 5.0 |
| $\delta$ – Weight for $\mathcal{L}_{cd}$ | 0.2 | 0.2 | 0.2 | 0.1 |
| $\eta$ – Relative strength of $\ell_1$-regularization | 0.5 | 0.5 | 1.0 | 3.0 |

Table 2: Hyperparameters for FLINT

Tab. 2 reports the setting of our hyperparameters for different datasets. We briefly discuss here our method to tune the different weights.

We varied $\gamma$ between 0.8 to 20 for all datasets, and stopped at a value for which the $\mathcal{L}_{if}$ loss seemed to optimize well (value dropped by at least 50% compared to the start). For MNIST and FashionMNIST,

| | $\eta = 1$ | $\eta = 2$ | $\eta = 3$ | $\eta = 5$ |
|---|---|---|---|---|
| no entropy | 92.7 | 90.4 | 91.2 | 84.2 |
| with entropy | 91.2 | 90.7 | 90.8 | 82.9 |

Table 3: Fidelity (in %) variation for $\eta$ and entropy losses for QuickDraw. $\delta = 0.1$ is fixed

the first value, 0.8 worked well. For the others, $\gamma$ needed to be increased so that the autoencoder worked well. Too high $\gamma$ might result in failed optimization due to exploding gradients.

The variation of $\beta$ was based on two indicators: (i) The system achieves high fidelity, for eg. at least 90%, so too small $\beta$ can't be chosen, (ii) For high $\beta$, the attributes become class-exclusive with only one attribute activating for a sample and result in high $\mathcal{L}_{if}$. Thus, $\beta$ was varied to get high fidelity and avoiding second scenario. $\beta = 0.5$ worked well for MNIST, FashionMNIST. For QuickDraw, we needed to decrease $\beta$ because of second scenario.

The system is fairly robust to choice of $\delta$, $\eta$. Too high $\ell_1$ regularization results in loss of fidelity (Tab. 3). These values were mostly heuristically chosen, and small changes to them do not cause much difference to training. We kept the effect of entropy low for ResNet because of its very deep architecture and high computational capacity of intermediate layers which can easily sway attributes to be class-exclusive.

### S.2.1.4   AM+PI procedure

In our case this optimization problem for an attribute $j$ is:

$$\arg \max_x \lambda_\phi \phi_j(x) - \lambda_{tv}\text{TV}(x) - \lambda_{bo}\text{Bo}(x)$$

where $\text{TV}(x)$ denotes total variation of $x$ and $\text{Bo}(x)$ promotes boundedness of $x$ in a range. We fix parameters for AM+PI for MNIST, FashionMNIST, QuickDraw as $\lambda_\phi = 2, \lambda_{tv} = 6, \lambda_{bo} = 10$ and $\lambda_\phi = 2, \lambda_{tv} = 20, \lambda_{bo} = 20$ for CIFAR10. For each sample $x_0$ to be analyzed, we analyze input for this optimization as $0.3x_0$ for MNIST, FashionMNIST, QuickDraw and as $0.4x_0$ for CIFAR10. For optimization, we use Adam with learning rate 0.05 for 300 iterations, halving learning rate every 50 iterations.

### S.2.1.5   Optimization and Runs

The models are trained for 12 epochs on MNIST, FashionMNIST and QuickDrawm and for 25 epochs on CIFAR-10. We use Adam [9] as the optimizer with fixed learning rate 0.0001 and train on a single NVIDIA-Tesla P100 GPU. Implementations are done using PyTorch [16].

**Number of runs**: For the accuracy and fidelity results in the main paper, we have reported mean and standard deviation for 4 runs with different seeds for each system. The conciseness results are computed by averaging conciseness of 3 models for each reported system.

### S.2.1.6   Resource consumption

Compared to $f$, $\Psi, h$ and $d$ have fewer parameters. For networks shown in Fig. 1, the LeNet based predictor has around 800,000 trainable parameters, interpreter $g$ contains 70,000 parameters, decoder $d$ contains 3000 parameters. For networks in Fig. 2, ResNet based predictor contains 11 million parameters, interpreter $g$ contains 530,000 parameters, and decoder $d$ contains 4.9 million parameters (almost all of them in the last FC layer). In terms of space, FLINT occupies more storage space according to the decoder, but is still of comparable size to that of only storing predictor.

**Training time**   In terms of training time consumption there is lesser difference when $f$ is a very deep network, due to all networks $\Psi, h, d$ being much shallower (lesser number of layers) than $f$. For eg. on both CIFAR-10, QuickDraw, FLINT consumes just around 10% more time for training compared to training just the predictor (BASE-$f$). The difference is more pronounced on with shallower $f$ where $\Psi, h, d$ also have comparable number of layers to $f$. Training BASE-$f$ on MNIST consumes 50% less time compared to FLINT.

We compare the average training times (for four runs) for SENN and FLINT in Tab. 4. Each model is trained for the same number of epochs, on the same computing machine (1 NVIDIA Tesla P100 GPU). It is clear that SENN requires significantly more time to train. This is primarily because of gradient of output w.r.t input being part of their loss function. Thus the computational graph for a forward pass is twice as big as their model architecture and followed by a backward pass through the bigger graph.

| Dataset | SENN | FLINT |
|---|---|---|
| MNIST | 2311 | **518** |
| FashionMNIST | 2333 | **519** |
| CIFAR-10 | 10210 | **1548** |
| QuickDraw | 10548 | **1207** |

Table 4: Training times for FLINT and SENN (in seconds)

### S.2.2 Experiments on CIFAR-100 and CUB-200

We also demonstrate the ability of the system to handle more complex datasets by experimenting with CIFAR100 [10] and Caltech-UCSD-200 (CUB-200) fine-grained Bird Classification dataset [20]. CIFAR100 contains 100 classes with 500 training and 100 testing samples per class (image size $32 \times 32 \times 3$). CUB-200 contains 11,788 images of 200 categories of birds, 5,994 for training and 5,794 for testing. We scale each sample in CUB-200 to size $224 \times 224 \times 3$. We also don't crop using the bounding boxes and use the full images for training and testing.

Compared to our earlier experiments, we make two key changes to the framework, (i) Increase size of dictionary of attribute functions to accommodate larger images/number of classes, (ii) Modify architecture of decoder $d$ with more upsampling and convolutional layers. For CIFAR100, the same architectures for $f$ and $g$ as on CIFAR10 is used, but with $J = 72$. We apply random horizontal flip as additional augmentation and train for 51 epochs. For CUB-200, we use the ResNet18 [7] for large-scale images as predictor architecture. We use $J = 180$, and apply random horizontal flip and random cropping of zero-padded image as data augmentation. The predictor is initialized with network pretrained on ImageNet and trained for 50 epochs. For both datasets, we do not vary the other hyperparameters much compared to experiments on CIFAR10. The hidden layers accessed are same for both. The hyperparameters of the interpretability loss remain unchanged for CIFAR100 and for CUB-200 we increase $\beta$ and $\gamma$ to $1.0$ and $3.0$, respectively.

We report the accuracy of BASE-$f$, FLINT-$f$ and FLINT-$g$ models (single run) and fidelity of FLINT-$g$ to FLINT-$f$ in Tab. 5 and conciseness below in Fig. 3. It should be noted that due to high number of classes, the disagreements between $f$ and $g$ are more common. The generated interpretations for the class predicted by $f$ can still be useful if it is among top classes predicted by $g$ (for a more detailed discussion, see Sec. S.2.3.2). Thus we report below top-$k$ fidelity of $g$ to $f$ for $k = 1, 5$ (the default fidelity of interpreter metric corresponds to $k = 1$). We also illustrate visualizations of sample relevant class-attribute pairs with global relevance $r_{j,c} > 0.5$ in Fig. 4 for CIFAR100, and for CUB-200 in Figs. 5, 7, 6 , 8.

**Key observations**: FLINT-$f$ achieves almost the same accuracy as BASE-$f$ model for both datasets, competitive for models of this size. Given the large number of classes, it achieves high fidelity of interpretations with top-1 fidelity of more than 80% and top-5 fidelity around 97% for both datasets. The effect of increased number of classes and complexity of datasets is also seen in comparatively higher conciseness of FLINT. However, relative to the total number of attributes, the interpretations still utilize small fraction of them, similar to results on other datasets. We also showcase the ability of

| | Accuracy (in %) | | | Fidelity (in %) | |
|---|---|---|---|---|---|
| Dataset | BASE-$f$ | FLINT-$f$ | FLINT-$g$ | Top-1 | Top-5 |
| CIFAR100 | 70.7 | 70.8 | 69.9 | 85.2 | 97.3 |
| CUB-200 | 71.3 | 71.0 | 68.7 | 80.0 | 96.7 |

Table 5: Results for accuracy (in %) and fidelity to FLINT-$f$ on CIFAR100, CUB-200.

attributes learnt in FLINT to capture interesting concepts. For eg. on CUB-200, we visualize various attributes which encode concepts like 'yellow-headed birds' (Fig. 5), 'red-headed birds' (Fig. 7), 'blue-faced birds' (Fig. 6) and 'long orange/red legs' (Fig. 8).

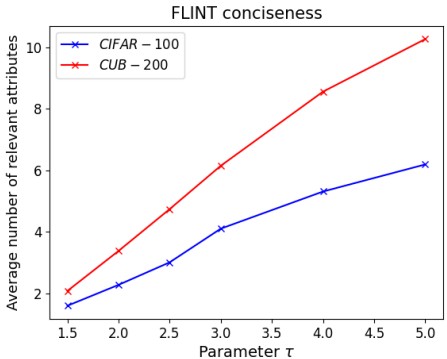

Figure 3: Conciseness curve of FLINT-$g$ interpretations on CIFAR100 and CUB-200

### S.2.3 Ablation Studies and Analysis

#### S.2.3.1 Shuffling experiment

By structure, for both FLINT-$g$ and SENN, the output are generated by combining high level attributes and weights. To test how crucial the learnt attributes are to their predictions, we shuffle the attribute values $\Phi(x)$ for each sample $x$ (this corresponds to shuffling $h(x)$ for SENN with their notations). This is an extreme test: we therefore expect an important drop in accuracy. Tab. 6 reports the results for the experiments for our method and SENN. More precisely, we calculate the drop in prediction accuracy of FLINT-$g$ (and SENN), compared to their mean accuracies. For SENN, the very small drop in accuracy indicates its robustness to this shuffling, which highlights the fact that in this model, the activation of a given subset of attributes is not crucial for the prediction. In contrast FLINT-$g$ relies strongly on its attributes for its prediction.

| Dataset | SENN | FLINT-$g$ |
|---|---|---|
| MNIST | 0.5 | 87.6 |
| FashionMNIST | 10.9 | 76.6 |
| CIFAR-10 | 17.5 | 74.4 |
| QuickDraw | 0.3 | 74.9 |

Table 6: FLINT and SENN accuracy drop for shuffled attributes (in %)

#### S.2.3.2 Disagreeement analysis

In this part, we analyse in detail the "disagreement" between the predictor $f$ and the interpreter $g$. Note that we already achieve very high fidelity to predictor for all datasets. We limit our analysis to QuickDraw, our dataset with least fidelity. Understanding disagreement can help us improving our framework as well as providing a measure of reliability about predictors output.

For a given sample with disagreement, if the class predicted by $f$ is among the top predicted classes of $g$, the disagreement is acceptable to some extent as the attributes can still potentially interpret the prediction of $f$. The worse kind of samples for disagreement are the ones where class predicted by $f$ is not among the top predicted classes of $g$, and even worse are where, in addition to this, $f$ predicts the true label. We thus compute the top-$k$ fidelity (for $k = 2, 3, 4$) on QuickDraw with ResNet, which for the default parameters described in the main paper, achieves a top-2 fidelity of $94.7\%$, top-3 fidelity $96.9\%$, and top-4 fidelity $98.2\%$. Only on $141$ (i.e. $0.7\%$) samples the class predicted by $f$, same as true class, is not in top-3 predicted by $g$ classes.

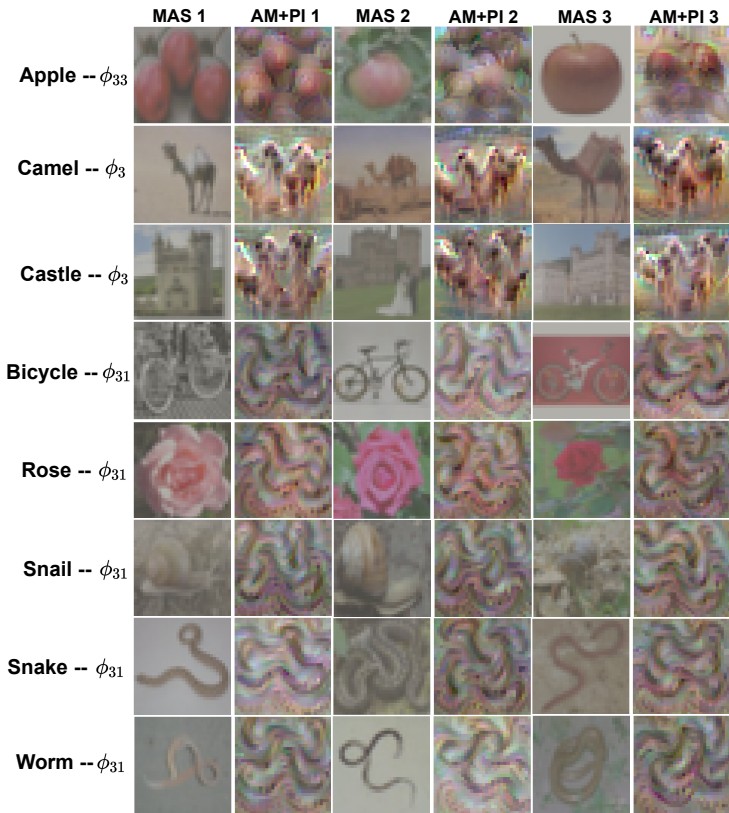

Figure 4: Sample class-attribute visualizations for CIFAR100. Three MAS and their corresponding AM+PI outputs are shown.

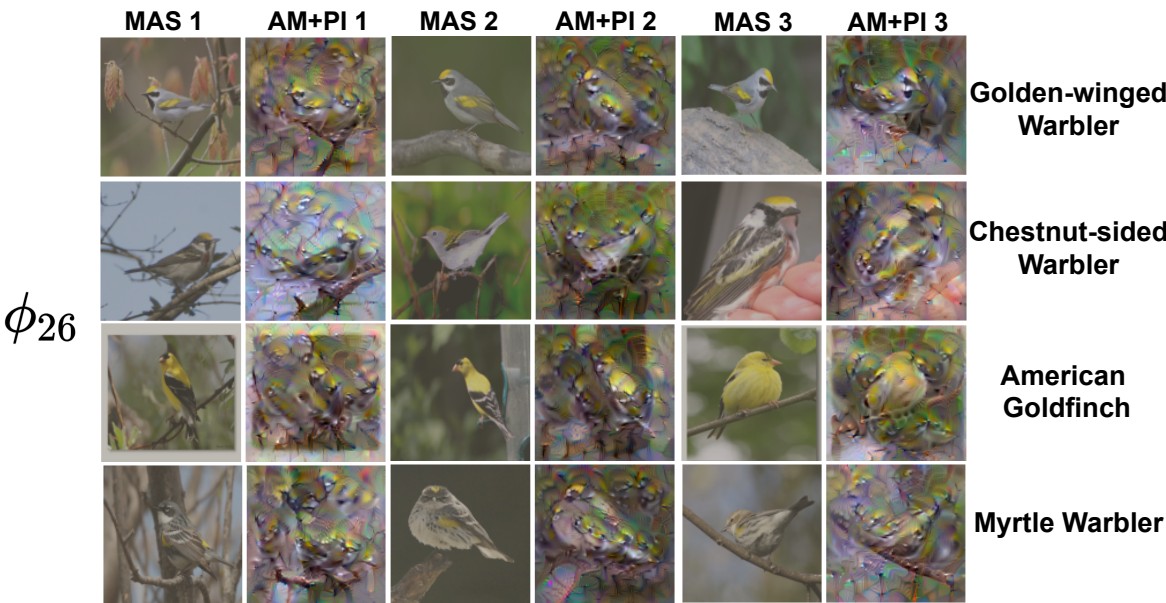

Figure 5: Relevant class-attribute pairs on CUB-200 with attribute $\phi_{26}$. Each row gives visualization for a relevant class of the attribute with three MAS and corresponding AM+PI outputs.

For eg., for the 'Apple' class (in QuickDraw), there only three disagreement samples for which $f$ delivers correct prediction (plotted in Fig. 9) are not resembling apples at all. We propose an

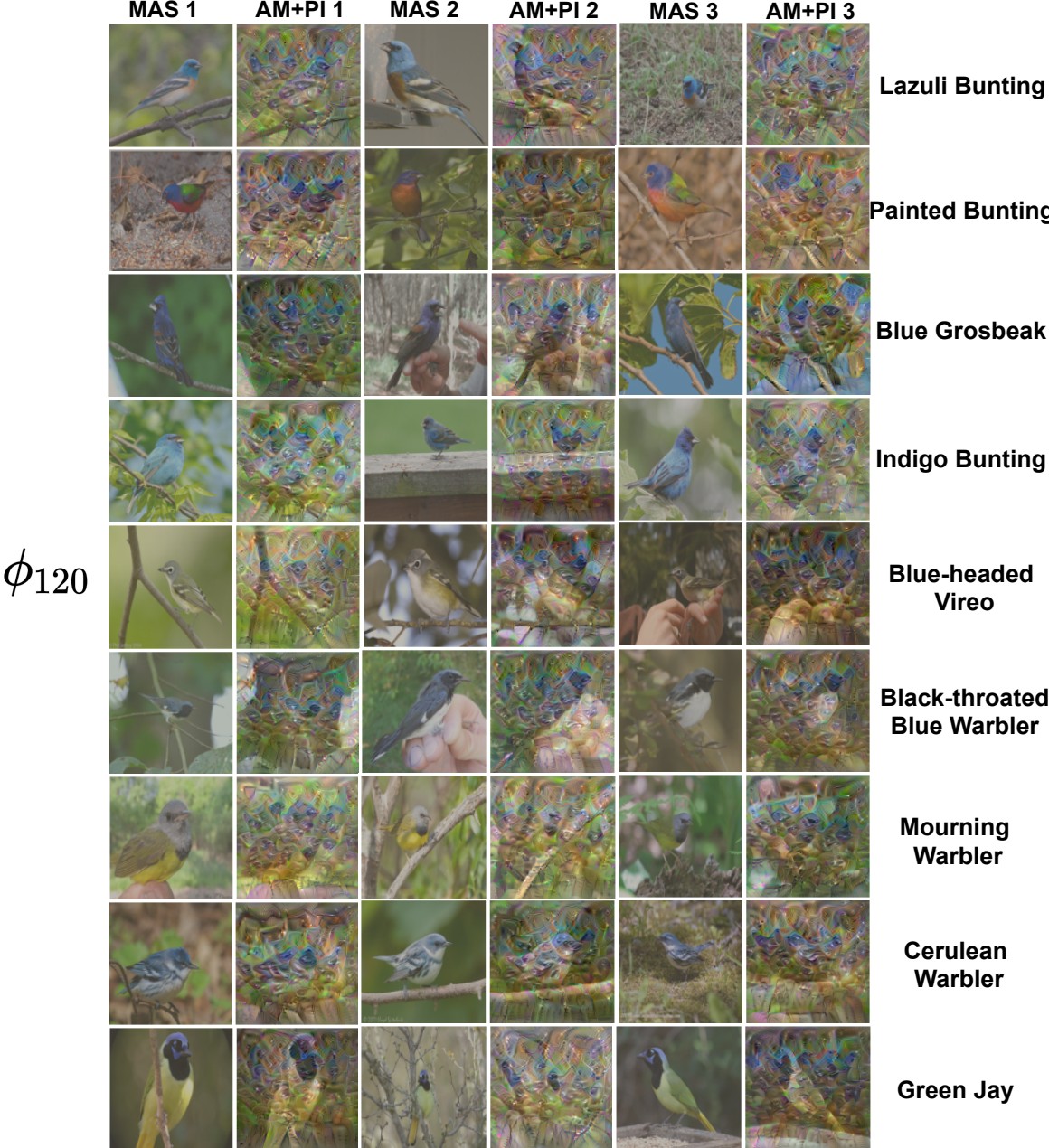

Figure 6: Relevant class-attribute pairs on CUB-200 with attribute $\phi_{120}$. Each row gives visualization for a relevant class of the attribute with three MAS and corresponding AM+PI outputs.

original analysis approach that consists in calculating a *robust centrality measure*—the projection depth—of these three samples as well as of another 100 training samples w.r.t. the 8000 training 'Apple' samples, plotted in Fig. 10. To that purpose, we use the notion of projection depth [24, 15] for a sample $\boldsymbol{x} \in \mathbb{R}^d$ w.r.t. a dataset $\boldsymbol{X}$ which is defined as follows:

$$D(\boldsymbol{x}|\boldsymbol{X}) = \left(1 + \sup_{\boldsymbol{p} \in \mathcal{S}^{d-1}} \frac{|\langle \boldsymbol{p}, \boldsymbol{x} \rangle - \mathrm{med}(\langle \boldsymbol{p}, \boldsymbol{X} \rangle)|}{\mathrm{MAD}(\langle \boldsymbol{p}, \boldsymbol{X} \rangle)}\right)^{-1}, \tag{1}$$

with $\langle \cdot, \cdot \rangle$ denoting scalar product (and thus $\langle \boldsymbol{p}, \boldsymbol{X} \rangle$ being a vector of projection of $\boldsymbol{X}$ on $\boldsymbol{p}$) and med and MAD being the univariate median and the median absolute deviation form the median. Fig. 10

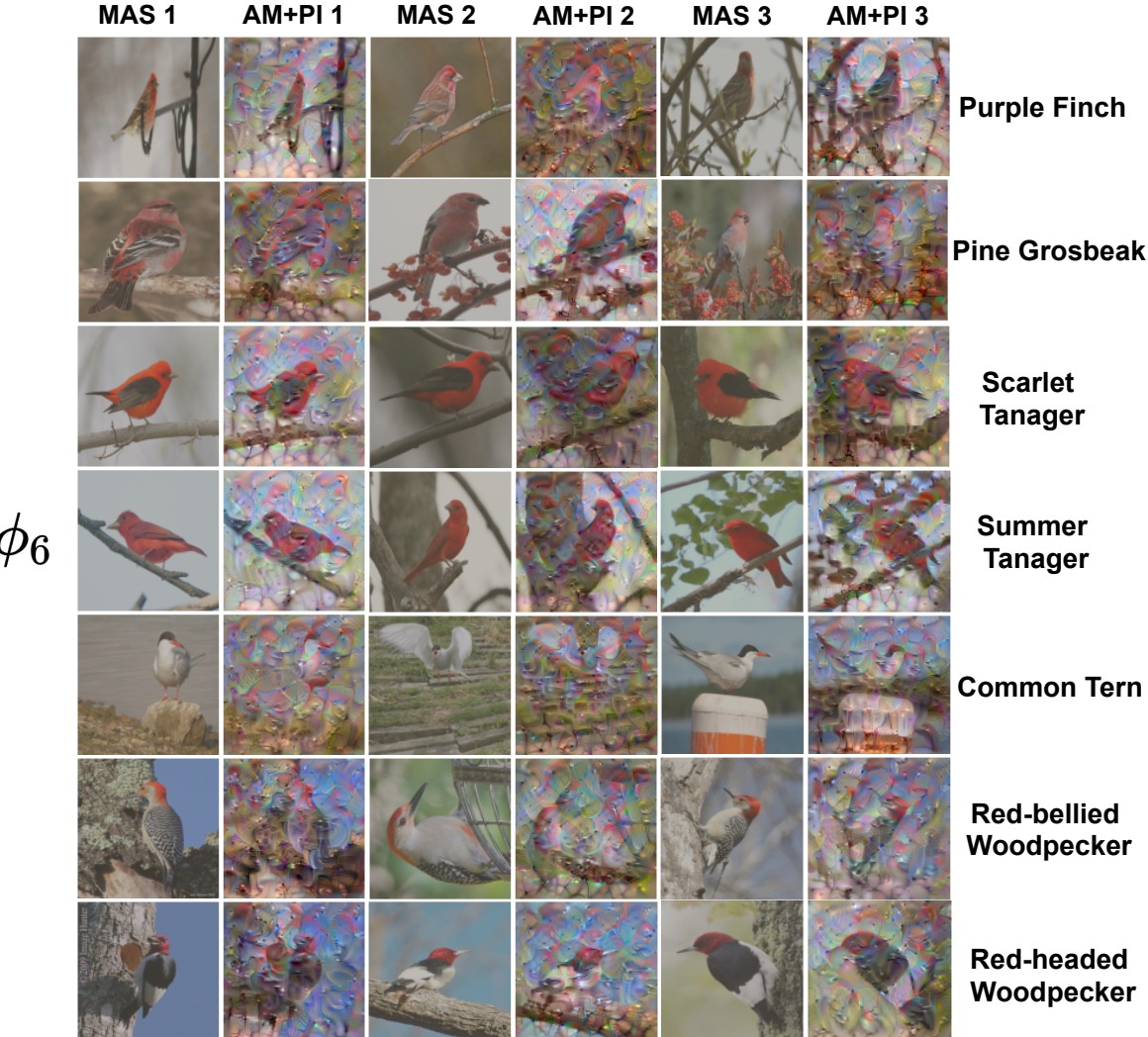

| MAS 1 | AM+PI 1 | MAS 2 | AM+PI 2 | MAS 3 | AM+PI 3 | |

$\phi_6$

Purple Finch
Pine Grosbeak
Scarlet Tanager
Summer Tanager
Common Tern
Red-bellied Woodpecker
Red-headed Woodpecker

Figure 7: Relevant class-attribute pairs on CUB-200 with attribute $\phi_6$. Each row gives visualization for a relevant class of the attribute with three MAS and corresponding AM+PI outputs.

confirms the visual impression that these 3 disagreement samples are outliers (since their depth in the training class is low).

Fig. 11 depicts 26 such cases for 'Cat' class to illustrate their logical dissimilarity. Being a complex model, the ResNet-based predictor $f$ still manages to learn to distinguish these cases (while $g$ does not), but in a way $g$ does not manage at all to explain. Eventually, exploiting disagreement of $f$ and $g$ could be used as a means to measure trustworthiness. Deepening this issue is left for future works.

### S.2.3.3 Effect of autoencoder loss

Although the effect of $\mathcal{L}_{of}, \mathcal{L}_{cd}$ can be objectively assessed to some extent, the effect of $\mathcal{L}_{if}$ can only be seen subjectively. If the model is trained with $\gamma = 0$, the attributes still demonstrate high overlap, nice conciseness. However, it becomes much harder to understand concepts encoded by them. For majority of attributes, MAS and the outputs of the analysis tools do not show any consistency of detected pattern. Some such attributes are depicted in Fig. 12 Such attributes are present even for the model trained with autoencoder, but are very few. We thus believe that autoencoder loss enforces a consistency in detected patterns for attributes. It does not necessarily guarantee semantic meaningfulness in attributes, however it's still beneficial for improving their understandability.

| MAS 1 | AM+PI 1 | MAS 2 | AM+PI 2 | MAS 3 | AM+PI 3 |
|---|---|---|---|---|---|

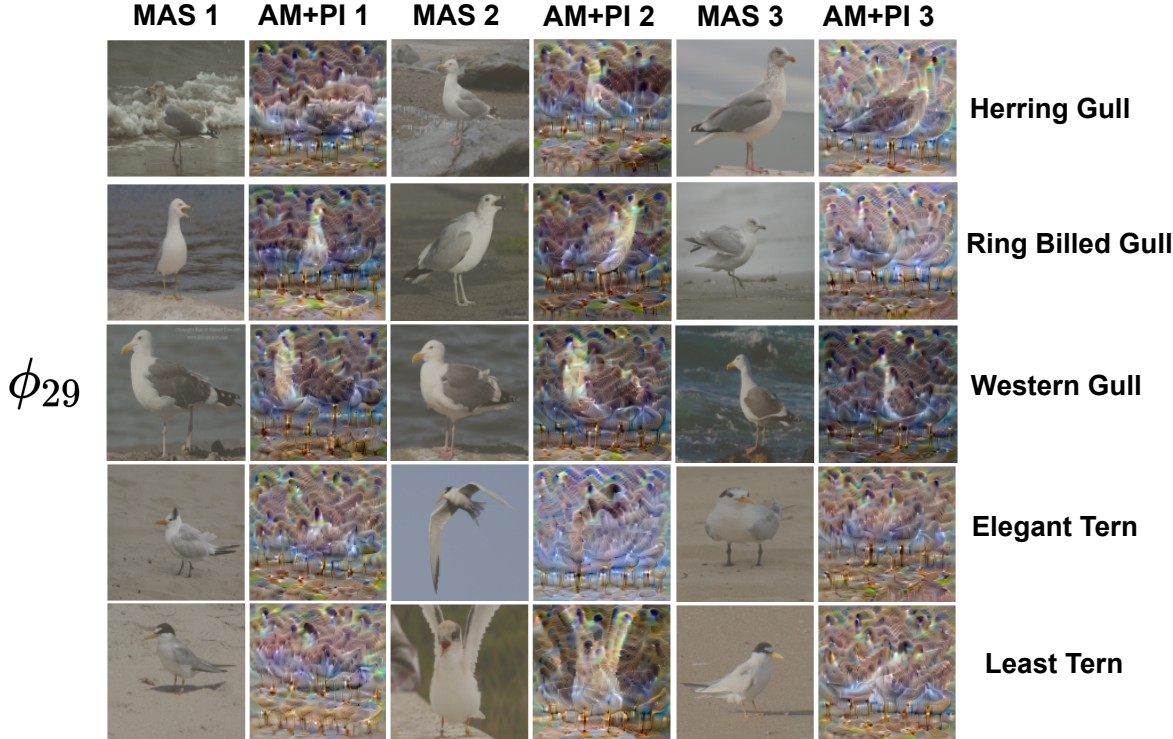

Figure 8: Relevant class-attribute pairs on CUB-200 with attribute $\phi_{29}$. Each row gives visualization for a relevant class of the attribute with three MAS and corresponding AM+PI outputs.

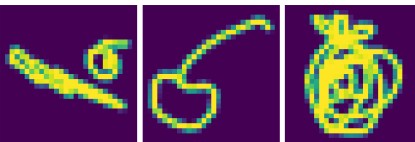

Figure 9: The three 'Apple' class samples classified correctly by $f$ but not by $g$.

#### S.2.3.4 Effect of hidden layer selection

We already discussed the empirical rationale behind our choice of hidden layers in Sec. S.2.1.1. In general for any predictor architecture or dataset, the most obvious choice is to select last convolutional layer output. This also helps achieving high fidelity for $g$. The only problem that might arise when selecting layer(s) very close to the output is that the attribute might be learnt trivially. This is indicated by extremely low entropy and high input fidelity loss. While tuning hyperparameters of interpretability loss could be helpful in tackling this issue (reducing $\beta$, increasing $\gamma$), choosing an earlier hidden layer can also prove to be very useful. We study the effect of choice of hidden layers with ResNet18 on QuickDraw. We make 3 different choices of single hidden layers (9th, 13th, 16th conv layers). For each choice we tabulate resulting metrics (accuracy, fidelity of interpreter, reconstruction loss, conciseness for threshold $1/\tau = 0.2$) in Tab. 7. All other hyperparameters remain same.

| Layer | Accuracy (in %) | Fidelity (in %) | $\mathcal{L}_{if}$ | Conciseness $1/\tau = 0.2$ |
|---|---|---|---|---|
| 9th conv | 85.2 | 78.0 | 0.074 | 1.873 |
| 13th conv | 85.6 | 85.6 | **0.073** | 1.905 |
| 16th conv | 86.5 | **96.0** | 0.081 | **1.562** |

Table 7: Effect of different hidden layers for Resnet18 on QuickDraw.

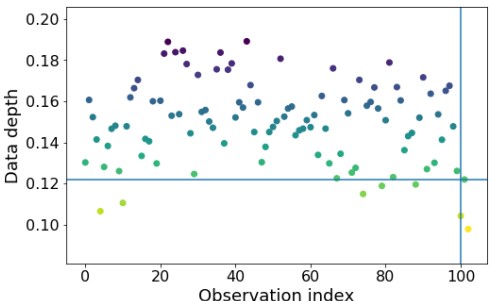

Figure 10: Projection data depth calculated with (1) w.r.t. the 8000 'Apple' training sample for 100 'Apple' test samples and for the three (observation indices 101–103) 'Apple' class samples classified correctly by $f$ but not by $g$.

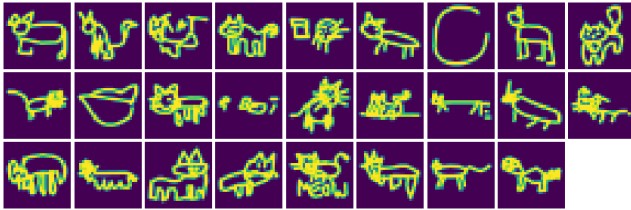

Figure 11: 26 samples from 'Cat' class which are not in top3 $f$-predicted classes.

**Key observations**: (a) Compared to average BASE-$f$ accuracy of 85.3% for ResNet18 on QuickDraw, accuracy of all models are comparable or slightly better. Thus, choice of hidden layers does not strongly affect predictor accuracy. (b) The interpreter fidelity gets considerably better if the layer chosen is closer to the output. (c) The input fidelity/reconstruction loss does not behave as monotonously, but it is not surprising that layers close to the output result in worse input reconstruction. (d) Interpretations are expected to be more concise when chosen layer is very close to the output in the sense that conciseness is an indicator of abstraction level of the interpretation. Thus, a standard choice is to start with a layer close to the output. A small revision may be needed depending upon optimization of input fidelity loss.

### S.2.3.5 Effect of number of attributes J

**Effect of $J$**   We study the effect of choosing small values for number of attributes $J$ (keeping all other hyperparameters same). Tab. 8 tabulates the values of input fidelity loss $\mathcal{L}_{if}$, output fidelity loss $\mathcal{L}_{of}$ on the training data by the end of training for MNIST and the fidelity of $g$ to $f$ on MNIST test data for different $J$ values. Tab. 9 tabulates same values for QuickDraw. The two tables clearly show that using small $J$ can harm the autoencoder and the fidelity of interpreter. Moreover, the system packs more information in each attribute and this makes it hard to understand them, specially for very small $J$. This is illustrated in Figs. 13 and 14, which depict part of global interpretations generated on MNIST for $J = 4$ (all the parameters take default values). Fig. 13 shows global class-attribute relevances and Fig. 14 shows generated interpretation for a sample attribute $\phi_2$. It can be clearly seen that the attributes start encoding concepts for too many classes (high number of bright spots). This also causes their AM+PI outputs to be muddled with two many patterns. This adds a lot of difficulty in understandability of these attributes.

|          | $\mathcal{L}_{if}$ (train) | $\mathcal{L}_{of}$ (train) | Fidelity (test) (%) |
|----------|----------------------------|----------------------------|---------------------|
| $J = 4$  | 0.058                      | 0.57                       | 87.4                |
| $J = 8$  | 0.053                      | 0.23                       | 97.5                |
| $J = 25$ | 0.029                      | 0.16                       | 98.8                |

Table 8: Effect of $J$ on losses and fidelity for MNIST with LeNet.

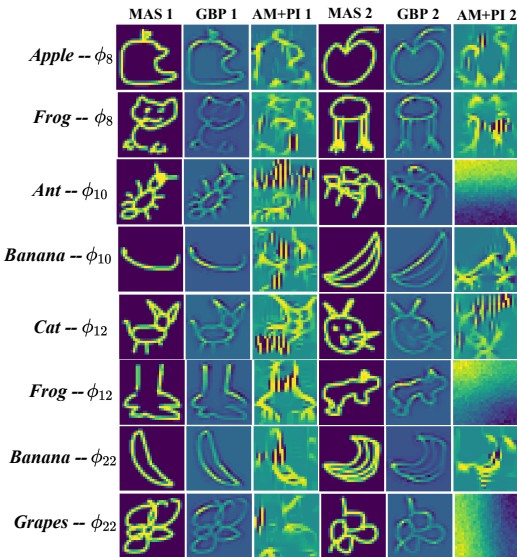

Figure 12: Sample class-attribute pair visualizations learnt without autoencoder loss $\mathcal{L}_{if}$. GBP stands for Guided Backpropagation.

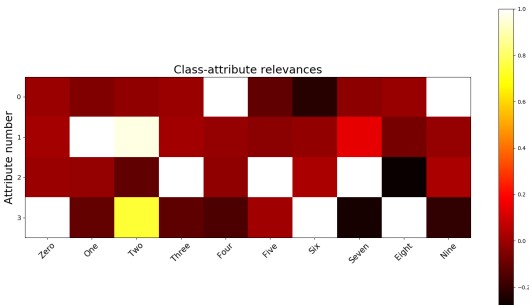

Figure 13: Global class attribute relevances for model with $J = 4$ on MNIST.

**How to choose the number of attributes** Assuming a suitable architecture for decoder $d$, simply tracking $\mathcal{L}_{if}, \mathcal{L}_{of}$ on training data can help rule out very small values of $J$ as they result in poorly trained decoder and relatively poor fidelity of $g$. One can also qualitatively analyze the generated explanations from the training data to tune $J$ to a certain extent. Too small values of $J$ can result in attributes encoding concepts for too many classes, which affects negatively their understandability. It is more tricky and subjective to tune $J$ once it becomes large enough so that $\mathcal{L}_{if}, \mathcal{L}_{of}$ are optimized well. The upper threshold of choosing $J$ is subjective and highly affected by how many attributes the user can keep a tab on or what fidelity user considers reasonable enough. It is possible that due to enforcement of conciseness, even for high value of $J$, only a small subset of attributes are relevant for interpretations. Nevertheless, for high $J$ value, there is a risk of ending up with too many attributes or class-attribute pairs to analyze.

|  | $\mathcal{L}_{if}$ (train) | $\mathcal{L}_{of}$ (train) | Fidelity (test) (%) |
|---|---|---|---|
| $J = 4$ | 0.094 | 2.08 | 19.5 |
| $J = 8$ | 0.079 | 1.48 | 57.6 |
| $J = 24$ | 0.069 | 0.34 | 90.8 |

Table 9: Effect of $J$ on losses and fidelity for QuickDraw with ResNet.

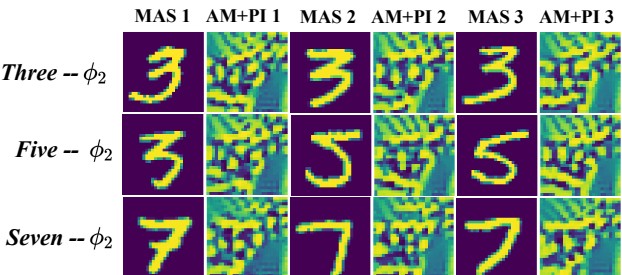

Figure 14: Interpretation for attribute $\phi_2$ for model learn on MNIST with $J = 4$.

It is important to notice that it is possible to select $J$ from the training set only by using a cross-validation strategy. In practise, it seems reasonable to agree on smallest value of $J$ for which the increase of the cross-validation fidelity estimate drops dramatically, since further increase of $J$ would generate less understandable attributes with very little gain in fidelity.

### S.2.3.6 Effect of loss scheduling

We also study the effect of introduction of different schedules for output fidelity and conciseness loss with ResNet18 on CIFAR10. We introduce $\mathcal{L}_{of}$ at different points of time during training (indicated by first column of Tab. 10. $\mathcal{L}_{cd}$ is introduced 1 epoch later. The first row corresponds to current setting proposed in the main paper. Total training time constitutes 25 epochs. All other hyperparameters remain same.

| Time of introduction | Accuracy (in %) | Fidelity (in %) | $\mathcal{L}_{if}$ | Conciseness $1/\tau = 0.2$ |
|---|---|---|---|---|
| Epoch 3 (current) | 84.6 | 93.5 | 0.421 | 2.612 |
| Epoch 4 | 84.8 | 93.4 | 0.427 | 2.501 |
| Epoch 5 | 84.3 | 94.2 | 0.426 | 2.351 |
| Epoch 6 | 85.0 | 93.1 | 0.426 | 2.376 |
| Epoch 8 | 84.5 | 93.7 | 0.432 | 2.642 |
| Epoch 10 | 84.6 | 93.9 | 0.422 | 1.944 |
| Epoch 14 | 84.2 | 92.1 | 0.445 | 2.274 |
| Epoch 21 | 84.6 | 91.2 | 0.450 | 3.710 |
| Epoch 24 | 84.4 | 86.3 | 0.524 | 4.533 |

Table 10: Effect of loss scheduling for Resnet18 on CIFAR10.

**Key Observations**: (a) As soon as the system receives reasonable time to train with all three losses (note that input fidelity loss is always present), small changes to introduction of losses have little to no impact on the metrics. (b) By contrast, when we introduce the losses extremely late (for eg. see the last two rows), the interpretability losses/metrics get noticeably worse.

### S.2.3.7 Additional visualizations

For completeness, we show some additional visualizations of global interpretations (relevances, class-attribute pairs) and local interpretations.

Fig. 15 contains global relevances generated for MNIST and FashionMNIST. Global relevances for QuickDraw and CIFAR10 are in main paper.

Figs. 16, 17, 18, 19 show some additional class-attribute pairs and their visualizations for all 4 datasets. Local interpretations on some test samples from these datasets are depicted in Figs. 20, 21, 22, 23.

### S.2.4 Other tools for analysis

Although we consider AM+PI as the primary tool for analyzing concepts encoded by attributes (for MAS of each class-attribute), other tools can also be helpful in deeper understanding of the attributes. We introduce two such tools:

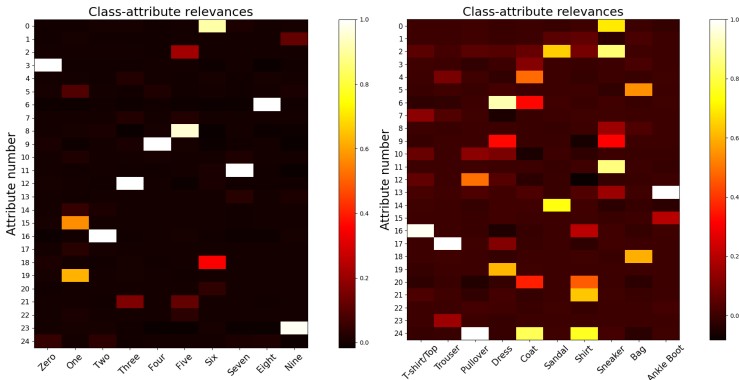

Figure 15: Global class-attribute relevances $r_{j,c}$ for MNIST (Left) and FashionMNIST (Right). 14 class-attribute pairs for MNIST and 26 pairs for FashionMNIST have relevance $r_{j,c} > 0.2$.

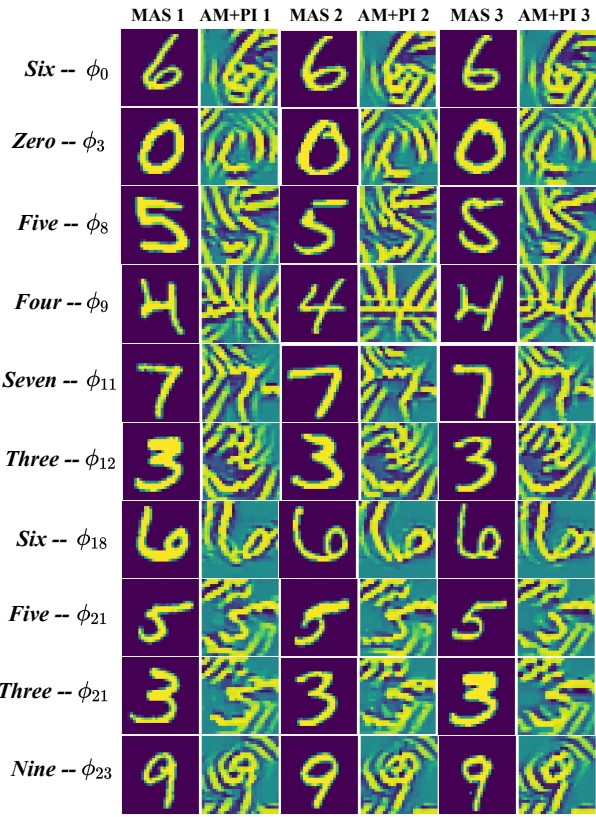

Figure 16: Additional class-attribute visualizations for MNIST. Three MAS and their corresponding AM+PI outputs are shown.

- *Input attribution*: This is a natural choice to understand an attribute's action for a sample. Any algorithms ranging from black-box local explainers to saliency maps can be employed. These maps are less noisy (compared to AM+PI) and very general choice, applicable to almost all domains.

- *Decoder*: Since we also train a decoder $d$ that uses the attributes as input. Thus, for an attribute $j$ and $x$, we can compare the reconstructed samples $d(\Phi(x))$ and $d(\Phi(x)\backslash j)$ where $\Phi(x)\backslash j$ denotes attribute vector with $\phi_j(x) = 0$, i.e., removing the effect of attribute $j$. While, the above comparison can be helpful in revealing information encoded in attribute $j$, it is not guaranteed to do so as the attributes can be entangled.

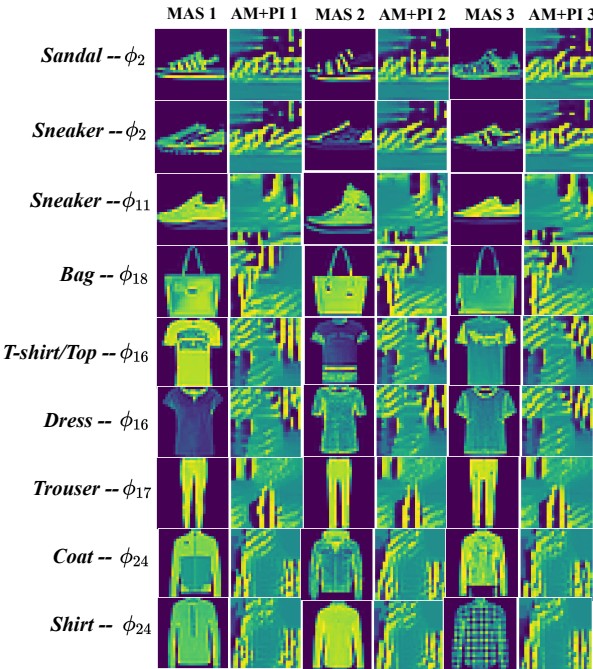

Figure 17: Additional class-attribute visualizations for Fashion-MNIST. Three MAS and their corresponding AM+PI outputs are shown.

We illustrate the use of these tools for certain example class-attribute pairs on QuickDraw in Fig. 24 and 25. Note that as discussed in the main paper, these tools are not guaranteed to be always insightful, but their use can help in some cases.

Fig. 24 depicts example class-attribute pairs where decoder $d$ contributes in understanding of attributes. The with $\phi_j$ column denotes the reconstructed sample $d(\Phi(x))$ for the maximum activating sample $x$ under consideration. The without $\phi_j$ column is the reconstructed sample $d(\Phi(x))\backslash j$ with the effect of attribute $\phi_j$ removed for the sample under consideration ($\phi_j(x) = 0$). For eg. $\phi_1, \phi_{23}$, strongly relevant for Cat class, detect similar patterns, primarily related to the face and ears of a cat. The decoder images suggest that $\phi_1$ very likely is more responsible for detecting the left ear of cat and $\phi_{23}$, the right ear. Similarly analyzing decoder images for $\phi_{22}$ in the third row reveals that it is likely has a preference for detecting heads present towards the right side of the image. This is certainly not the primary concept $\phi_{22}$ detects as it mainly detects blotted textures, but it certainly carries information about head location to the decoder.

Fig. 25 depicts example class-attribute pairs where input attribution contributes in understanding of attributes. We use Guided Backpropagation [19] (GBP) as input attribution method for ResNet on QuickDraw. It mainly assists in adding more support to our previously developed understanding of attributes. For eg., analyzing $\phi_5$ (relevant for Dog, Lion) based on AM+PI outputs suggested that it mainly detects curves similar to dog ears. The GBP output support this understanding as the most salient regions of the map correspond to curves similar to dog ears.

### S.2.5  Baseline implementations

We cover the implementation details of various baselines used in this work (Tab 2, 3, 4 from main paper). As stated in the main paper, implementation of our method is available on Github [1].The accuracy of FLINT-$f$ is compared against BASE-$f$, PrototypeDNN, SENN. Fidelity of FLINT-$g$ is compared against VIBI and LIME.

---

[1]`https://github.com/jayneelparekh/FLINT`

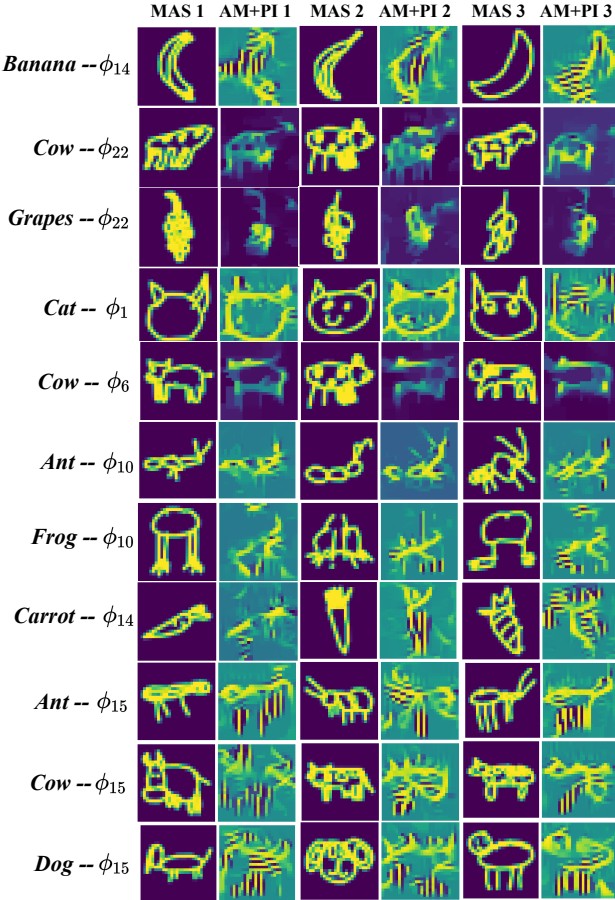

Figure 18: Additional class-attribute visualizations for QuickDraw. Three MAS and their corresponding AM+PI outputs are shown.

**BASE-$f$**    We compare accuracy of FLINT-$f$ with BASE-$f$. The BASE-$f$ model has the same architecture as FLINT-$f$ but is trained with $\beta, \gamma, \delta = 0$, that is, only with the loss $\mathcal{L}_{pred}$ and not interpretability loss term. All the experimental settings while training this model are same as FLINT.

**PrototypeDNN**    We directly report the accuracy of PrototypeDNN on MNIST, FashionMNIST (Tab 2 main paper) from the results mentioned in their paper [13]. Note that we do not report any results of PrototypeDNN on CIFAR10 and QuickDraw. This is because for processing more complex images and achieving higher accuracy, one would need to non-trivially modify architecture of their proposed model. Thus to avoid any unfair comparison, we did not report this result. The results of BASE-$f$ and SENN on CIFAR, QuickDraw help validate performance of FLINT-$f$ on QuickDraw.

**SENN**    We compare the accuracy as well as conciseness curve for FLINT with Self-Explaining Neural Networks (SENN) [2]. We implemented it with the help of their official implementation available on GitHub [2]. SENN employs a LeNet styled network for MNIST in their paper. We use the same architecture for MNIST and FashionMNIST. For QuickDraw and CIFAR10 we use the VGG based architecture proposed for SENN in their paper to process more complex images. However, to maintain fairness, the number of attributes used in all the experiments for SENN are same as those for FLINT, that is, 25 for MNIST & FashionMNIST, 24 for QuickDraw and 36 for CIFAR10, and also train for the same number of epochs. We use the default choices in their implementation for all hyperparameters and other settings. Another notable point is that although interpretations of SENN are worse than FLINT in conciseness (even when compared non-entropy version of FLINT), the

---

[2] https://github.com/dmelis/SENN

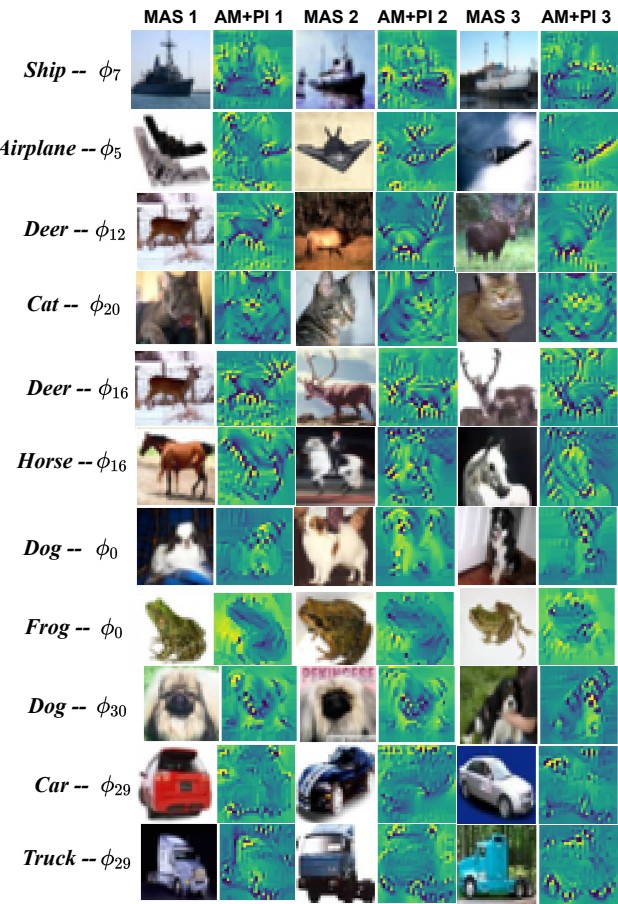

Figure 19: Additional class-attribute visualizations for CIFAR-10. Three MAS and their corresponding AM+PI outputs are shown.

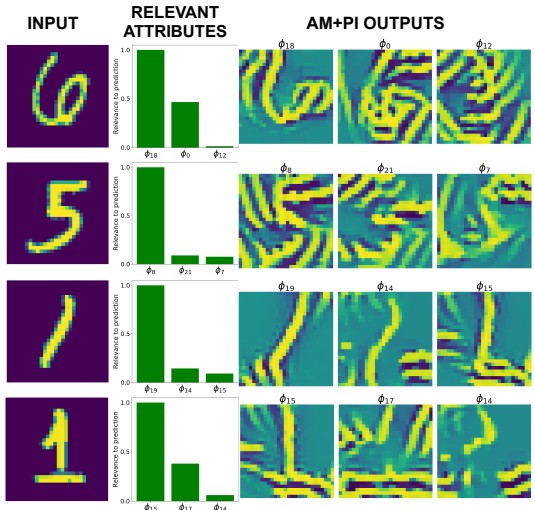

Figure 20: Local interpretations on test samples for MNIST. True labels are: 'Six', 'Five', 'One' and 'One'. Top 3 most relevant attributes and their corresponding AM+PI outputs are shown.

strength of $\ell_1$ regularization in SENN is 2.56 times our strength (for identical $\mathcal{L}_{pred}$, i.e, cross-entropy loss with weight 1.0).

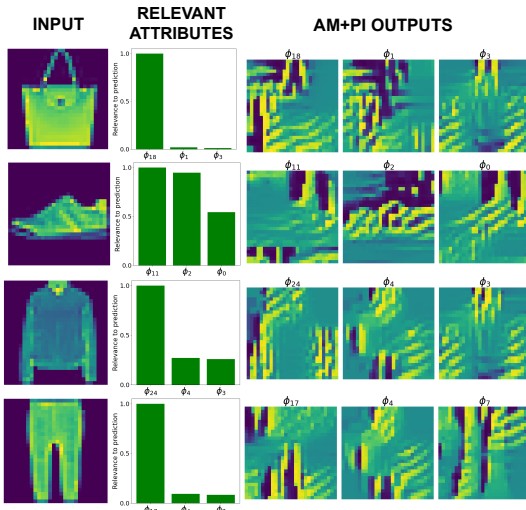

Figure 21: Local interpretations on test samples for Fashion-MNIST. True labels are: 'Bag', 'Sneaker, 'Coat', 'Trousers'. Top 3 most relevant attributes and their corresponding AM+PI outputs are shown.

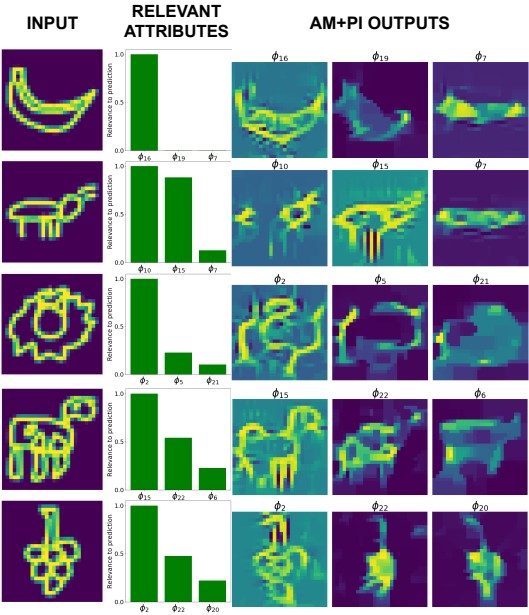

Figure 22: Local interpretations on test samples for QuickDraw. True labels are: 'Banana', 'Ant', 'Lion', 'Cow' and 'Grapes'. Top 3 most relevant attributes and their corresponding AM+PI outputs are shown.

**VIBI & LIME** We benchmark the fidelity of interpretations of FLINT-$g$ for both by-design and post-hoc interpretation applications against a state-of-the-art black box explainer variational information bottleneck for interpretation (VIBI) [3] and traditional explainer LIME [17]. Note that VIBI also possesses a model approximating the predictor for all samples. Both methods are implemented using the official repository for VIBI [3]. We compute the "*Approximator Fidelity*" metric as described in their paper, for both systems. In the case of VIBI, this metric exactly coincides with our definition of fidelity. We set the hyperparameters to the setting that yielded best fidelity for datasets reported in their paper. For VIBI, chunk size $4 \times 4$, number of chunks $k = 20$, for LIME, chunk size $2 \times 2$, number of chunks $k = 40$. The other hyperparameters were the default parameters in their code.

---

[3] https://github.com/SeojinBang/VIBI

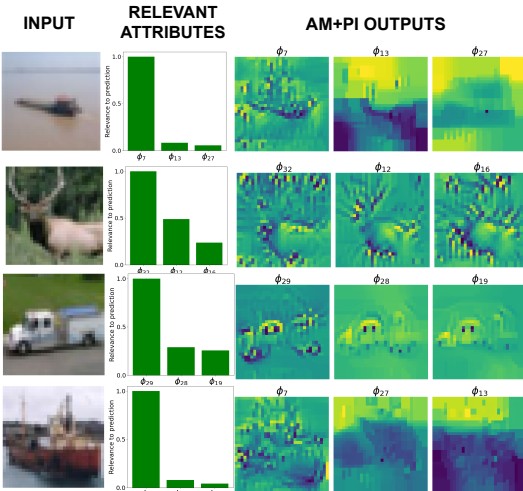

Figure 23: Local interpretations on test samples for CIFAR-10. True labels are: 'Ship', 'Deer', 'Truck' and 'Ship'. Top 3 most relevant attributes and their corresponding AM+PI outputs are shown.

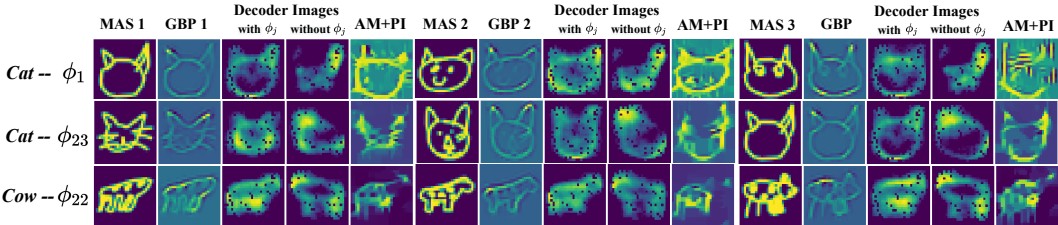

Figure 24: Examples of class-attribute pairs on QuickDraw, where decoder assists in understanding of encoded concept for the attribute.

### S.2.6 Subjective evaluation details

The form taken by the participants can be accessed here [4]. 17 of the 20 respondents were in the age range 24-31 and at least 16 had completed a minimum of masters level of education in fields strongly related to computer science, electrical engineering or statistics. The form consists of a description where the participants are briefly explained through an example the various information (class-attribute pair visualizations and textual description) they are shown and the response they are supposed to report for each attribute, which is the level of agreement/disagreement with the statement: "The patterns depicted in AM + PI outputs can be meaningfully associated to the textual description". As mentioned in the main paper, four descriptions (questions #2, #5, #8, #9 in the form) were manually corrupted to better ensure that participants are informed about their responses. The corruption mainly consisted of referring to other parts or concepts regarding the relevant class which are *not* emphasized in the AM+PI outputs.

## S.3 Post-hoc interpretations

### S.3.1 Implementation details

The network architecture, the optimization procedures and hyperparameters are set to exactly the same values they were for their 'by-design', with one small change, $\beta$ for CIFAR10 is used as 0.3, and not 0.6, this is because for $\beta = 0.6$, the system was running into scenario discussed in Sec. S.2.1.3, thus $\beta$ was lowered.

---

[4] https://forms.gle/PW6DEPZSmXb46Lnv9

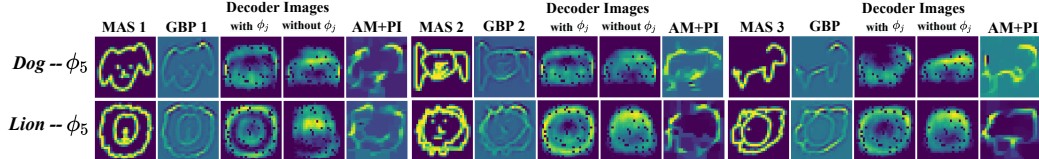

| | MAS 1 | GBP 1 | Decoder Images with $\phi_j$ without $\phi_j$ | | AM+PI | MAS 2 | GBP 2 | Decoder Images with $\phi_j$ without $\phi_j$ | | AM+PI | MAS 3 | GBP | Decoder Images with $\phi_j$ without $\phi_j$ | | AM+PI |

Figure 25: Examples of class-attribute pairs on QuickDraw, where input attribution (GBP) assists in understanding of encoded concept for the attribute. GBP stands for Guided Backpropagation.

| Dataset | VIBI | FLINT-$g$ |
|---|---|---|
| MNIST | 95.8±0.2 | **98.6±0.2** |
| FashionMNIST | 88.4±0.2 | **92.8±0.3** |
| CIFAR10 | 64.2±0.3 | **89.1±0.5** |
| QuickDraw | 78.0±0.4 | **90.5±0.3** |

Table 11: Fidelity for post-hoc interpretations of BASE-$f$ (in %)

**Results.** Fidelity benchmarked against VIBI is tabulated in Tab. 11 and conciseness curves for post-hoc interpretations are shown in Fig. 26. They clearly indicate that FLINT can yield high fidelity and highly concise *post-hoc* interpretations.

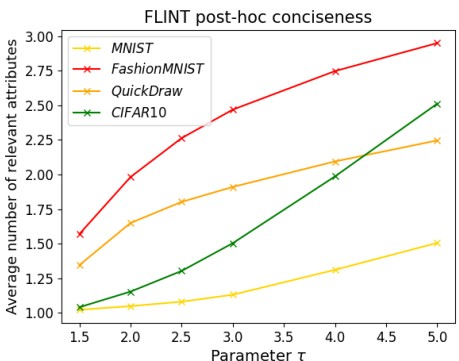

Figure 26: Conciseness curve of post-hoc interpretations generated using FLINT

## S.3.2 Additional visualizations

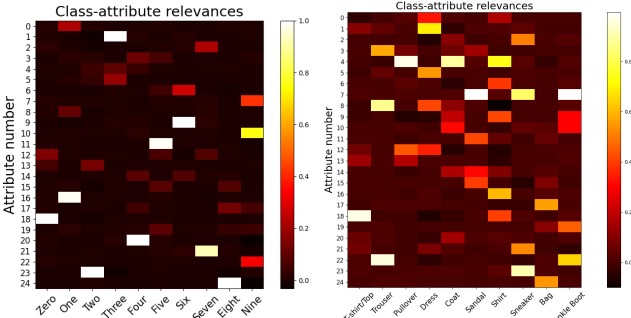

Figure 27: Global class-attribute relevances $r_{j,c}$ for post-hoc interpretations on MNIST (Left) and FashionMNIST (Right). 15 class-attribute pairs for MNIST and 28 pairs for FashionMNIST have relevance $r_{j,c} > 0.2$.

Figs. 27 and 28 contain global relevances for post-hoc interpretations on all four datasets. Figs. 29, 30, 31 and 32, illustrate some additional visualizations of class-attribute pairs on all datasets.

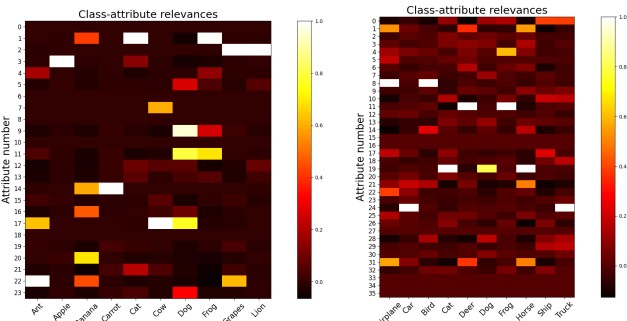

Figure 28: Global class-attribute relevances $r_{j,c}$ for post-hoc interpretations on QuickDraw (Left) and CIFAR10 (Right). 24 class-attribute pairs for QuickDraw and 26 pairs for CIFAR10 have relevance $r_{j,c} > 0.2$.

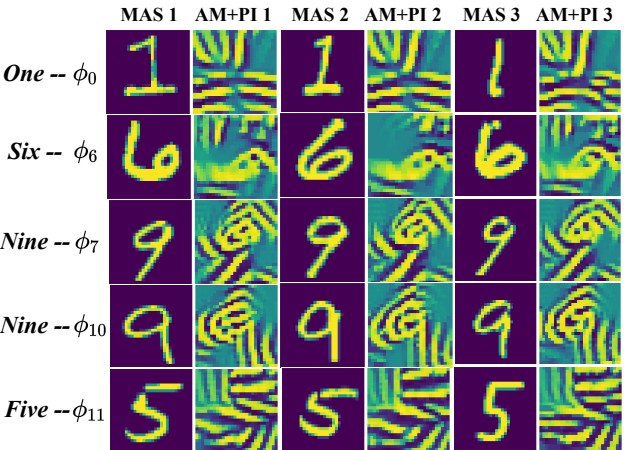

Figure 29: Sample class-attribute visualizations for post-hoc interpretations for MNIST.

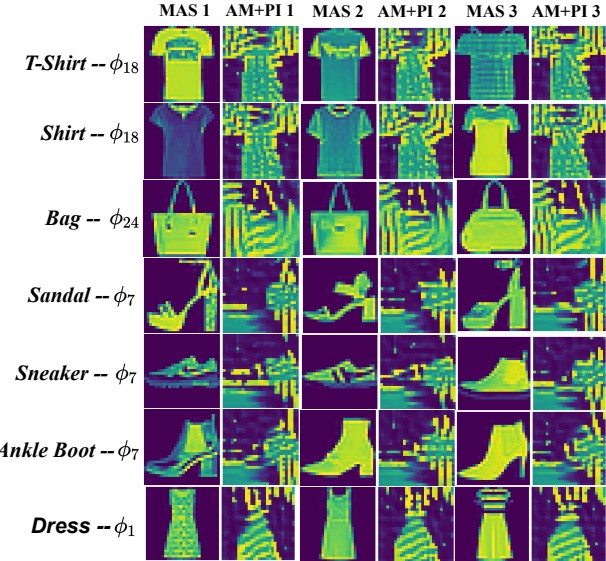

Figure 30: Sample class-attribute visualizations for post-hoc interpretations for Fashion-MNIST

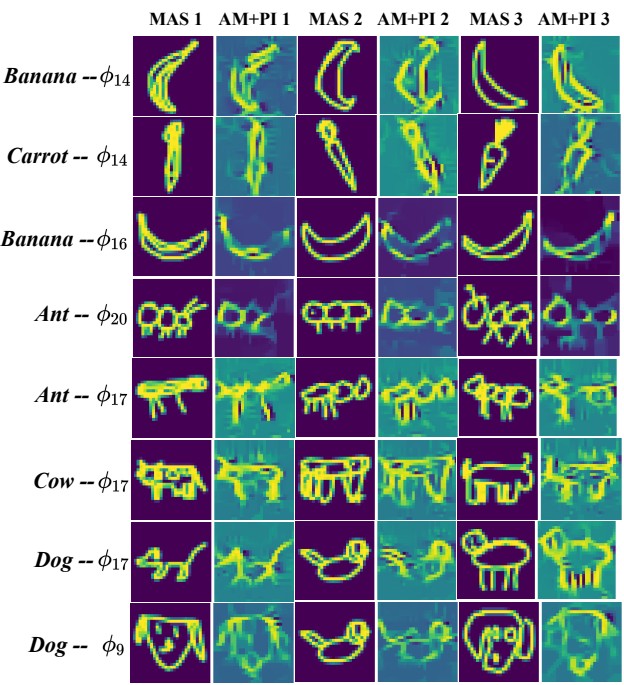

Figure 31: Sample class-attribute visualizations for post-hoc interpretations on QuickDraw

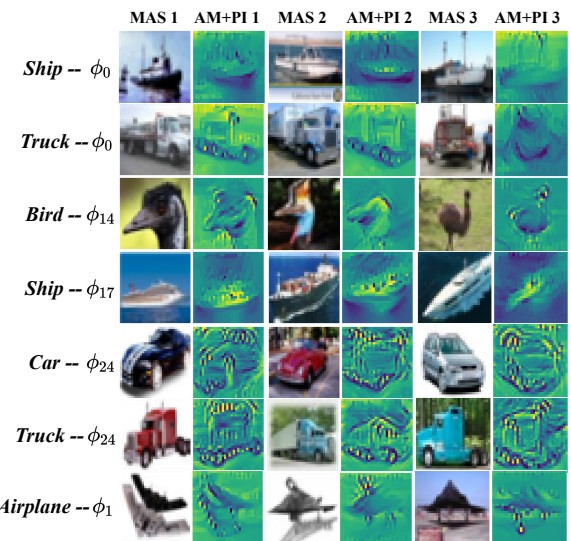

Figure 32: Sample class-attribute visualizations for post-hoc interpretations on CIFAR-10

### S.3.3 Experiments using ACE

We conducted additional experiments using ACE to interpret trained models from our experiments. The key bottleneck for ACE's application on our datasets and networks is the use of CNN as a similarity metric (to automate human annotation) for image segments irrespective of their scale, aspect ratio. This is a specialized property only been empirically shown for specific CNN's trained on ImageNet (as discussed in their paper). The networks trained on our datasets thus very often cluster unrelated segments, resulting in little to no consistency in any extracted concept. To illustrate the above we describe the experimental settings and show extracted concepts for a few classes from QuickDraw and CIFAR-10 on the BASE-$f$ models. The quality of results is the same when interpreting FLINT-$f$ models although we only illustrate interpretations from BASE-$f$ models.

**Experimental setting.**   We utilize the official open-sourced implementation of their method [5]. Due to the smaller sized images we perform segmentation at a single scale. We experimented with different configurations for "number of segments" and "number of clusters/concepts". The number of segments were varied from 3 to 15. For higher values the segments were often too small for concepts to be meaningful. We thus kept the number of segments 5 for each sample. For each class we chose 100 samples. The number of clusters were varied from 5 to 25. Due to the smaller number of segments (compared to original experiments from ACE which used 25), we kept number of clusters at 12. We access the deepest intermediate layer used in experiments with FLINT (shown in Fig. 2).

**Results.**   The top 3 discovered concepts (according to the TCAV scores) are shown in Fig. 33. The segments for any concept on CIFAR show almost no consistency. This is mainly because the second step pf ACE, requiring a CNN's intermediate representations to replace a human subject for measuring the similarity of superpixels/segments, is hard to expect for these networks not trained on ImageNet. Thus, segments capturing background or any random part of the object, completely unrelated, end up clustered together. For QuickDraw, the segmentation algorithm also suffers problems in extracting meaningful segments due to sparse grayscale images. It generally extracts empty spaces or a big chunk of the object itself. This, compounded with the earlier issue about segment similarity results in mostly meaningless concepts. The only slight exception to this is concept 3 for 'Ant' for which two segments capture a single flat blob with small tentacles.

## S.4   Limitations

- The current design of attributes and their encoded concept visualization procedure is more suited for classification tasks and image as input modality. Although multiple proposed losses/visualization tools could be generalized to other input modalities (e.g. audio, video, graphs etc.) or other machine learning tasks (regression), it requires work in that direction.
- The set of proposed properties is not exhaustive and can be further improved. It could be desired that attributes encode concepts which are invariant to certain transformations, or focus on specific spatial regions, or are robust to adversarial attacks / specific types of noise or contamination.
- The choice of hidden layers requires some level of experience with neural architectures.

## S.5   Potential negative societal impact

Interpretability becoming a frequently raised issue when training and exploiting neural network (NN) architectures, the main expected societal impact of FLINT is improvement of their understandability as well as providing explanations of the decisions made by NNs. Nevertheless, even this intrinsically benevolent machinery can be used for harm when in malicious hands.

Potential misuse can be expected on two different levels: First, if incorrectly trained (e.g., wrong NN design, insufficient number of training examples and/or or training epochs, in particular for FLINT-$f$), due to lack of knowledge or on purpose, FLINT can provide misleading interpretations. Second, even a well-trained explainable AI can serve evil purpose in hands of a maliciously destined user.

Clearly, the authors expect proper use of the developed FLINT methodology, although direct misuse-protection mechanisms were not developed in this piece of research, not being the initial goal.