# OpenReview forum: "A Framework to Learn with Interpretation"
_NeurIPS.cc/2021/Conference — NeurIPS 2021 Poster_

### Official Review · Reviewer_gNz2 · 2021-07-16

**Rating:** 7
**Confidence:** 3

**Summary:**

The paper proposes FLINT a Framework to Learn with Interpretation to solve Supervised Learning with Interpretation task.
FLINT involves training 3 models on a preditor model an interpreter and a decoder. This framework provides both global and local interpretations.

The interpreter network takes as an input the output of different hidden layers from the predictor network then computes an attribute dictionary of functions phi g(x)= softmax(Wphi(x)). Different phi corresponds to different concepts  phi_j(x)= Psi_j (f_I(x)) where Psi_j is shallow network and f_I(x) is the output of hidden state I.  So The model parameters of g depend on the parameters of Psi and Weights. Given interpreter with parameter and input x the contribution of phi_j = phi_j(x)*w_j,y_hat/max|ph_i*w_i_yhat| where y_hat the class output.

Global Interpretation is a set of class-attribute pairs (c, ph_j) such that their global relevance is greater than some threshold.
Local interpretation for a sample x provided is the set of attribute functions ph_j  with a local relevance score greater than some threshold.

FLINT has multiple loss functions, first to ensure that the interpreter output is similar to that of the model output cross-entropy is used. An additional loss is used to ensure diversity of ph_j and ensure input fidelity the input is reconstructed from phi using a decoder network which gives us the final loss.

FlINT was evaluated on 4 different image datasets, it was compared with other self explain networks SENN and PrototypeDNN to terms of accuracy and saliency methods LIME and VIBI in terms of fidelity.

**Limitations And Societal Impact:**

Yes, in the supplementary.

**Main Review:**

Strength:

- Although self-explaining networks have been previously proposed FLINT framework itself is original and novel.
- The paper is technically sound and the claims are well supported.
- The paper organized and is well written.
- The relation to previous work is clearly outlined.
- The empirical results are quite impressive epically that FLINT_g  produces accuracy similar to that of FLINT_f.
- The paper included a subject evaluation showing that the concepts learned by FLINT are understandable by humans.
- The proposed method is significant, the constraints used to impose Fidelity to Output, Conciseness and Diversity of Interpretations, and Fidelity to Input can be adapted to many methods.



Weakness:

- It is unclear how one can choose the hidden layer to give input to the interpreter.
- Comparison was done on very simple datasets, it is unclear if this will generalize to larger more complicated datasets.
- Baseline models used have very low accuracy compared to state-of-the-art accuracy.
- Empirical evaluation is somewhat limited.

Questions and Comments:

- Accuracy reported on CIFAR10 as an example has baseline accuracy 84.7 and FLINT was able to maintain this however state of the art accuracy is around 98% if that model was used as a baseline or other high performing network will FLINT be able to maintain such accuracy?
- The paper only used very simple datasets which are generally easy to train, how will this approach work on larger datasets like image net will the proposed optimization still work.
- It would be interesting to see how  FLINT compares to other self-explaining networks such as INVASE [1] or FRESH[2].



[1] Yoon, Jinsung, James Jordon, and Mihaela van der Schaar. "INVASE: Instance-wise variable selection using neural networks." International Conference on Learning Representations. 2018.
[2] Jain, Sarthak, et al. "Learning to faithfully rationalize by construction." arXiv preprint arXiv:2005.00115 (2020).


Overall, I believe the proposed method is great and I really enjoyed reviewing the paper.

**Time Spent Reviewing:**

5.5

---

> ### Author Response · Authors · 2021-08-10
> **Response to Reviewer gNz2**
>
> We appreciate the reviewer’s positive feedback and are happy to address their questions/comments. We respond to the three questions/comments in the first three points and then address the how to choose hidden layers in the fourth point:
>
>
>
> 1. Our primary purpose was to highlight the ability of FLINT on a relevant baseline. Nothing prevents applying FLINT approach to more complex network. The flexibility in choosing an earlier layer or decreasing strength of interpretability loss term can assist in maintaining high-level of accuracy. Moreover, additional strategies like data augmentation, large training epochs, optimization scheduling, etc., often used for improving accuracy can also be exploited without hindering FLINT. For example, we repeated the experiment on CIFAR-10 during the rebuttal period with just an additional augmentation of random horizontal flip and training for $N_{epochs}=75$. Simply updating the augmentation strategy and keeping $N_{epochs}=25$ same caused $\approx$ 2\% increase for both BASE-$f$ and FLINT-$f$. With $N_{epochs}=75$, the best accuracy for BASE-$f$ and FLINT-$f$ models were $89.3$, $89.2$\% respectively. Thus, around $4.7$\% increase. This was obtained without losing out on the interpretability losses.
>
>
>
>
> 2.  We do expect the proposed optimization to work on more complex datasets. Experimenting with more complex datasets warrants **at least** two modifications when applying FLINT: (1) Updating the decoder $d$ architecture accordingly and, (2) Increasing the number of attributes $J$ to accommodate larger number of classes and/or images. To illustrate this we experimented with CIFAR-100 during the rebuttal period. We made the following changes from the CIFAR-10 experiments: (1) Increasing $J = 72$, (2) Adding batchnorm layers to $f$ for better accuracy, (3) Updating architecture for $d$, (4) Training for 50 epochs. Note that all other hyperparameters and choices remained the same. We achieved an accuracy $\approx$ 71\% (for both FLINT-$f$ and BASE-$f$). From the point of view of interpretation, we achieve fidelity (or Top-1 fidelity) of 85.2\%, Top-5 fidelity of 97.3\%, with slightly better autoencoder performance and slightly worse conciseness compared to CIFAR-10. Note that ResNet-18 typically achieves around 75\% accuracy (https://github.com/weiaicunzai/pytorch-cifar100) but with significantly more training time (200 epochs) and learning rate schedulers.
>
>
>
> 3. Our focus was more on selecting baselines with similar means of explanations as ours. SENN is the closest, followed by PrototypeDNN. FRESH and INVASE are both interesting works (we can update our references with them) that generate explanations by performing feature selection over the raw input features. FRESH seems to be specifically designed for text processing applications. Like with the other interpretable by-design methods (except SENN), only accuracy as a metric is comparable with FLINT. It is much harder to conduct comparison on other aspects. Nevertheless, it points to a deeper, interesting research issue of devising evaluation strategies (objective/subjective) that can effectively compare interpretability of methods with different means of interpretation (for example, logical rules vs feature attribution).
>
>
>
> 4. **Choice of hidden layers**: We discuss about this in supplementary Sec 2.1.1. We can create a separate subsection in supplements for this, if needed.
>
>     For any dataset/model, the most standard choice of choosing the hidden layer, is to start with layers close to the output as they are expected to capture higher order features [1]. A common choice being to select last convolutional layer output. This also helps achieving high fidelity for $g$. This choice worked well for LeNet based network. As mentioned in L50-52 of supplementary, one potential optimization issue can arise when choosing a layer too close to the output - which is that attributes are learnt trivially. The interpreter achieves very high fidelity but very poor reconstruction loss ($L_{if}$), with only one attribute activating per sample. We faced this issue with ResNet18 model and it was hard to tackle this by tuning loss hyperparameters. To improve reconstruction loss we selected an earlier layer (13th layer specifically). It resulted in a decent input reconstruction loss but lower fidelity. Thus we also included a higher layer (15th) for better output fidelity without losing much on reconstruction.
>
>     In summary, choosing last convolutional, or an equivalent layer for higher order representations, is a very standard option to start. This can significantly reduce effort in making a reasonable choice, regardless of the task/model. One may need to slightly adapt this choice to ensure satisfactory optimization of $L_{if}$ if that is not the case. Since adapting this for a given model requires user to have some understanding of the architecture itself, we had included this choice as one of our limitations.
>
>     References: [1] Zeiler, M. D. and Fergus, R. Visualizing and understanding convolutional networks. ECCV 2014

---

> ### Comment · Reviewer_gNz2 · 2021-08-26
> **Thank you**
>
> I would like to thank the author for their response. They have addressed my concerns. I believe the method is novel and the overall quality of the work is high. My score remains as is.

---

### Official Review · Reviewer_ZpKd · 2021-07-16

**Rating:** 6
**Confidence:** 4

**Summary:**

The paper proposes FLINT, a new learning schema in which a neural net and a gray-box model are learned jointly by minimizing a composite loss: a loss on the labels + a set of losses controlling the concepts acquired by the gray-box models +
a distillation loss that encourages the gray-box models to behave similarly to the neural net.  The concepts in the gray-box model are built from intermediate neurons of the neural net to encourage faithfulness.  The authors also describe a visualization pipeline and a post-hoc explanation protocol.

**Ethical Concerns:**

None.

**Limitations And Societal Impact:**

The limitations are barely discussed in the introduction; there is a section on limitations and negative impact in the supplementary.

**Main Review:**

The paper is generally well written;  all concepts are explained clearly.  The text is structured just fine.  I do not have any major gripes with the presentation itself.

One issue is that the message is unfocused: it is not exactly clear what the authors want to say with this paper.  Apparently it is only about "going beyond SENNs", but I think that there are some interesting and deeper issues implicit in this work that the authors do not address:

    1.  There is an unstated benefit of gray-box models like SENNs that the authors overlook:  SENNs are not only interpretable by design but also *faithful* by design.  This is no longer the case in FLINT: there is no guarantee that the g network is 100% faithful to the f network.  This also explains why SENNs underperform in Table 1: they trade off performance for interpretability AND faithfulness.  This actually highlights a limitation of FLINT and that this should be explicitly discussed in the paper.

    2.  This also means that, implicitly at lesat, the authors suggest that it is fine to give up on faithfulness -- that faithfulenss is not always fundamental.  I don't think that it is reasonable to do so in high-stakes applications.  In order to defend this point, the authors neglect to clarify which applications their method is best suited for.

The core idea -- namely that of training two twin networks based on (roughly speaking) the same base intermediate layers but with different aims -- is sensible.  Re-using the intermediate layer to tie the two models is an interesting choice and is reminiscent of multi-task learning solutions.  Again, this introduces a performance-faithfulness trade-off that should be addressed.  Some modeling choices (like combining the L1 norm with entropy) are quite empirical, but on the bright side they seem to perform well in practice.

The experimental validation is a bit limited, but the choice of competitors and research questions is reasonable.  The results seem to favor the proposed method.


As for the score, I am a bit on the fence.  All in all, the contribution is sensible and the experiments are reasonable, but the paper is somewhat lacking in terms of focus, significance and depth.

**Time Spent Reviewing:**

6

---

> ### Author Response · Authors · 2021-08-10
> **Response to Reviewer ZpKd**
>
> We thank the reviewer for the careful reading and insightful comments.
>
> This work originated from discussions with practitioners of deep neural networks who wished to keep the accuracy of dedicated (convolutional) architectures developed for supervised classification while benefiting from interpretations of the decision. FLINT proposes interpretations that rely on (some) hidden layers of the prediction network, ensuring some level of faithfulness of the interpreter network with respect to the predictor network. Overall, FLINT is a general framework which allows to learn a pair of models by minimizing a set of criteria (input and output fidelity, diversity, conciseness) which are not meant here to be exhaustive but already cover a large number of desirable properties.
>
> 1. Faithfulness is an important feature and highly desirable. We note that for self-explained networks as well as for natural language processing, a few authors have already discussed faithfulness (see paper about SENN) and proposed to measure it through indirect criteria such as feature removal and/or stability. However the definition of faithfulness of an interpretation regarding a decision process has not yet reached a consensus particularly in the case of post-hoc interpretability or when the two models, predictor and interpreter, differ [Yin et al. 2021]. Other notions like explanation plausibility may also be of some relevance (see [Jacovi and Goldberg]). We think that measuring faithfulness of FLINT as well as imposing it during learning (i.e. going beyond fidelity) deserves a full additional paper. However we can propose a faithful solution by-design keeping the same framework (see point 2).
>
> 2. Now with our current framework FLINT, if faithfulness is regarded as the primary objective, we advise to take the interpreter network as the ultimate decision network as suggested in the conclusion. In this case, there is only one network and the so-called prediction network has played the useful role of  providing relevant hidden layers.
>
>
> To conclude, in this paper, we propose to add a paragraph about faithfulness and include these comments. We still would like to leave for further work the study of faithfulness with respect to all other criteria (fidelity, accuracy, confidence). We think that even partial faithfulness is acceptable if the user is aware when the interpretation is not faithful.
>
>
> References:
>
> [Jacovi and Goldberg] Alon Jacovi, Yoav Goldberg:
> Towards Faithfully Interpretable NLP Systems: How Should We Define and Evaluate Faithfulness? ACL 2020: 4198-4205.
>
> [Yin et al. 2021] Yin, Fan, Zhouxing Shi, Cho-Jui Hsieh, and Kai-Wei Chang. "On the Faithfulness Measurements for Model Interpretations." arXiv preprint:2104.08782.

---

> > ### Comment · Reviewer_ZpKd · 2021-08-30
> > **Response.**
> >
> > Thank you for your response.  I have no further questions.

---

### Official Review · Reviewer_eCgH · 2021-07-18

**Rating:** 6
**Confidence:** 2

**Summary:**

The paper introduces FLINT, a framework that jointly trains a classifier with an “interpreter”, an interpretable version of the model. The interpreter is a linear classifier over a set of attributes (each associated with a “concept”), which are derived from hidden states of the neural network. The interpreter is trained with three objectives: (1) matching the output of the original classifier, (2) a sparsity regularizer, and (3) an anto-encoding objective to reconstruct the input. Each attribute in the interpreter can be visualized by finding input that maximizes the attribute (with some regularization), which allows the interpreter to produce global interpretation for a class and local interpretation for an example. The method is tested on four image classification tasks. It has higher accuracy than two previous interpretable models (SENN and PrototypeDNN) and higher fidelity than two post hoc interpretation methods (LIME and VIBI). According to a human survey, the visualizations are human interpretable in most cases.

**Limitations And Societal Impact:**

I do not have any suggestions for this part.

**Main Review:**

Strength:
The framework is novel, intuitive, and flexible. It allows us to provide an interpretable version of any neural classifier. The experiments are extensive, and the results seem good. The model can retain high accuracy while producing high-fidelity interpretation.

Weakness:
My biggest concern is that the framework introduces extra complexity to model training. For example, the paper acknowledges that optimizing the full objective is hard, so they need a schedule to gradually increase the complexity of the objective (Section 3.1). This schedule may need to be changed for a new model. The framework also introduces several hyperparameters: the weight of different objectives and the number of attributes. These hyperparameters need to be tuned separately for different tasks/datasets.

In summary, I am leaning towards accepting the paper. However, I am not very familiar with the latest interpretation methods and how to evaluate them, so I might miss something.

**Time Spent Reviewing:**

3

---

> ### Author Response · Authors · 2021-08-10
> **Response to Reviewer eCgH**
>
> We thank the reviewer for their positive feedback and comments.
>
>
> The motivation for scheduling of the losses in Algorithm 1 is discussed in L216-217 of the main paper. The key reason behind the scheduling was simply to avoid introducing all three losses at the start. For instance, when the model is untrained, immediately introducing $L_{cd}$ loss potentially risks pushing all attributes to 0 (depending upon $l_1$ regularization strength). Further, to train with all three losses for majority of its training epochs, both losses $L_{cd}, L_{of}$ were introduced by the fourth epoch. We don't expect this choice to have any major impact on a new dataset. In practice, any schedule with a small delay in introducing $L_{of}, L_{cd}$ should work normally (for example, introducing both at third epoch).
>
>
> Sec 2.1.4, 2.1.3 from supplementary discuss the various factors involved in tuning the weights for different objectives and number of attributes. It is mainly driven by proper optimization for $L_{of}, L_{if}$. Although the framework can function meaningfully under a broad selection of hyperparameter values, it is true that they should be tuned according to the task/dataset. It is worth noting that, this overhead of selecting multiple hyperparameters is a common theme in other similar methods.

---

### Official Review · Reviewer_WJT2 · 2021-07-19

**Rating:** 6
**Confidence:** 5

**Summary:**

This paper proposes an architecture which jointly learns to classify as well as provide an explanation behind the classification. The explanation is generated from a separate part of the model which doesn’t affect the model performance much as the classification performance is achieved by a backbone network. The explanation generation part takes inputs from the intermediate layers of the backbone network which helps to produce better and more meaningful explanations. The results are promising on experiments shown on MNIST, FMNIST, CIFAR10 and QuickDraw datasets.

**Limitations And Societal Impact:**

Not in the main paper, as suggested in paper submission guidelines (provided in the supplementary)

**Main Review:**

Positives:

+ The overall idea of integrating explanations while learning to classify is important and timely, especially considering the spate of post-hoc explanation methods available today.
+ The use of an exclusive backbone network for classification helps maintain good classification performance.
+ Post-hoc explanation generation techniques can be applied in the framework.
+ The results show that the method performs better in terms of classification performance and explanation generation as well compared to other baseline methods.

Concerns:

- One of the primary concerns is whether the method is applicable to complex large-scale datasets, on which post-hoc attribution methods show their results (e.g. ImageNet). The paper does not show results on large-scale datasets, limiting the understanding of its applicability. As much as the proposed idea is interesting and new, to a certain extent, it attempts to improve over a couple of earlier methods in this direction such as SENN (Ref [4] in the paper). Showing it works on large-scale datasets would be one important improvement to show, especially considering the pieces used in the method are known before.

- The contribution of every attribute is given by the coefficients w which are given by the matrix W (considering all the classes) and trained as a simple fully connected layer. Once the model is trained, these contributions are fixed and are used as it is during inference. This is different for SENN, as the contributions are given by a relevance function there (realized by a neural network) and hence produces different contributions for different inputs. How do you view this difference? Do you think the same contributions for every input across all data points is justified?

- The interpreter selects outputs from one or more intermediate layers from the backbone network specified by the function f. The layers chosen for different architectures are shown in Fig.1 and Fig.2 of supplementary section. Are these choices arbitrary? If not, how were the layers chosen? I’m curious to know if this choice affects the quality of the interpretation. Also, if this is non-trivial and empirical, this makes the framework difficult to scale to newer datasets/models.

- In L56, it is said that ‘FLINT can be specialized to post-hoc interpretability if a pre-trained deep neural network is available.’ As far as I understood, the post-hoc interpretability is generated from the backbone network given by f, which is part of the whole network. Then, what is meant by availability of a pre-trained deep neural network here, and how is this achieved?

- The final loss function is an aggregation of many different loss functions which makes this network highly dynamical in nature. One needs to take proper care while selecting the coefficients and the number of epochs where every loss function was introduced. This was pointed out in Algorithm 1, but how were these epoch numbers arrived at? How does one choose these on a dataset?

- Table 1 compares the accuracy numbers with SENN and the proposed method uses a similar architecture as SENN for MNIST. However, for CIFAR10, SENN used smaller architectures for their results, which were not considered why. Was there any reason for this? Would this be fair comparison? I’m convinced with the explanations provided for the training time differences with SENN though.

- In L106-108, the paper talks about other methods which use a known dictionary of concepts (such as concept bottleneck models). Was there a particular reason not to compare with such methods? While it may be understandable, it may be nice for the paper to explicitly state why the baselines used in the experiments were chosen (and other such models were not).


**Time Spent Reviewing:**

4

---

> ### Author Response · Authors · 2021-08-10
> **Response to WJT2**
>
> We thank the reviewer for their feedback and are happy to respond to their concerns:
>
> 1. FLINT is primarily designed for learning interpretable networks by design. Thus, while selecting the datasets we experimented with those commonly used by other similar baselines.
>
>    Experimenting with more complex datasets warrants **at least** two modifications when applying FLINT: (1) Updating the decoder $d$ architecture accordingly and, (2) Increasing the number of attributes $J$ to accommodate larger number of classes and/or images. To illustrate this we experimented with CIFAR-100 during the rebuttal period. We made the following changes from the CIFAR-10 experiments: (1) Increasing $J = 72$, (2) Adding batchnorm layers to $f$ for better accuracy, (3) Updating architecture for $d$, (4) Training for 50 epochs. Note that all other hyperparameters and choices remained the same. We achieved an accuracy $\approx$ 71\% (for both FLINT-$f$ and BASE-$f$). From the point of view of interpretation, we achieve fidelity (or Top-1 fidelity) of 85.2\%, Top-5 fidelity of 97.3\%, with slightly better autoencoder performance and slightly worse conciseness compared to CIFAR-10. Note that ResNet-18 typically achieves around 75\% accuracy (https://github.com/weiaicunzai/pytorch-cifar100) but with significantly more training time (200 epochs) and learning rate schedulers.
>
>
> 2. Please note that the notion of relevance/contribution of attributes in SENN and FLINT are different. While the coefficients w, given by matrix W are fixed during inference in FLINT, the relevance/contribution score of an attribute for a sample is different for different inputs. We explain both these points in greater detail below:
>
>    **Relevance in FLINT**: FLINT defines two notions of relevance or contribution of an attribute: local and global (L166-175, main paper). Local relevance corresponds to relevance/contribution of an attribute $j$ for a given sample $x$. It is defined by first computing contribution of attribute $j$ to unnormalized score of predicted class $\phi_j(x).w_{j, \hat{y}}$, then normalizing this w.r.t all attributes. For an attribute to have a strong relevance/contribution in prediction of a sample, it requires **both**, a high coefficient $w_{j, \hat{y}}$, and strong detection of its associated concept given by *high activation on the given sample* $\phi_j(x)$. This is the reason why relevance/contribution for a sample in FLINT depends upon input even though $W$ is fixed during inference.
>
>    **Relevance in SENN**: SENN does not define any global relevance of a concept. By default relevance of a concept in SENN corresponds to local relevance for a given sample. It is defined precisely how reviewer WJT2 has stated, by the concept's weight corresponding to the predicted class, given by a coefficient network.
>
>    **Fixed W during inference**: Achieving better classification performance is a primary reason for modelling coefficients as function of input. However, as we show, performance (accuracy) of FLINT-$g$ is consistently competitive with SENN, justifying the current design. Moreover, we believe fixed W during inference is more favorable for interpretability. This is because the user is not further required to interpret/understand the function determining W. Nevertheless, nothing stops W to be modeled as output of a function during inference. More generally, the function $h$ could be modeled as some other interpretable function.
>
>
>
> 3. **Choice of hidden layers**: We discuss about this in supplementary Sec 2.1.1. We can create a separate subsection in supplements for this, if needed.
>
>     For any dataset/model, the most standard choice of choosing the hidden layer, is to start with layers close to the output as they are expected to capture higher order features [1]. A common choice being to select last convolutional layer output. This also helps achieving high fidelity for $g$. This choice worked well for LeNet based network. As mentioned in L50-52 of supplementary, one potential optimization issue can arise when choosing a layer too close to the output - which is that attributes are learnt trivially. The interpreter achieves very high fidelity but very poor reconstruction loss ($L_{if}$), with only one attribute activating per sample. We faced this issue with ResNet18 model and it was hard to tackle this by tuning loss hyperparameters. To improve reconstruction loss we selected an earlier layer (13th layer specifically). It resulted in a decent input reconstruction loss but lower fidelity. Thus we also included a higher layer (15th) for better output fidelity without losing much on reconstruction.
>
>
>
>     In summary, choosing last convolutional, or an equivalent layer for higher order representations, is a very standard option to start. This can significantly reduce effort in making a reasonable choice, regardless of the task/model. One may need to slightly adapt this choice to ensure satisfactory optimization of $L_{if}$ if that is not the case. Since adapting this for a given model requires user to have some understanding of the architecture itself, we had included this choice as one of our limitations.
>
>     References: [1] Zeiler, M. D. and Fergus, R. Visualizing and understanding convolutional networks. ECCV 2014
>
>
>
> 4. **Post-hoc interpretation** refers to the problem when a user wishes to interpret decisions of a given fixed model $\hat{f}$. Although FLINT is primarily designed to learn interpretable-by design networks, FLINT can also be applied to generate post-hoc interpretations for a given network $\hat{f}$ (supplied by the user) by fixing the predictor $f=\hat{f}$. For instance, any off-the-shelf neural network for image classification would fit this role. Only the parameters $\theta_\Psi, \theta_h, \theta_d$ are then learnt, through the interpretability loss (L352-354, main paper). In our experiments for post-hoc interpretations (Sec 6, main paper), $\hat{f}$ was chosen as LeNet network for MNIST and Fashion MNIST and ResNet18 network for CIFAR-10 and QuickDraw, each of them trained exclusively for accuracy.
>
>
> 5. The motivation for scheduling of the losses in Algorithm 1 is mentioned in L216-217 of the main paper. When the model is untrained, immediately introducing $L_{cd}$ loss potentially risks pushing all attributes to 0 (depending upon $l_1$ regularization strength). Otherwise, the time of introducing $L_{cd}$ doesn't have significant effect on optimization of other losses. Introducing the $L_{of}$ loss at the very start of training when the predictor's output is random and the attributes don't have any structure sometimes lead to low fidelity or sub-optimal performance of the interpreter. Since most models at least attain a small level of accuracy quickly, introducing $L_{of}$ after a small number of epochs like 1-3 should work normally. We have also not seen any advantage in delaying introduction of $L_{of}$ much later.
>
>
>
>    Thus, the key reason behind the scheduling was simply to avoid introducing all three losses at the start. To train the system with all three losses for majority of its training epochs, both losses $L_{cd}, L_{of}$ were introduced by the fourth epoch. We **don't** expect this choice to have any major impact on a new dataset. In practice, any schedule with a small delay in introducing $L_{of}, L_{cd}$ should work normally (for example. introducing both at third epoch).
>
>
>
> 6. The architecture for coefficient network for SENN reported in their paper is based on VGG11. This is also what we have reported results on. Their official implementation also includes possibility to use VGG13, VGG16, VGG19 based architectures. Due to the large training times we only experimented with VGG11 \& VGG13, and VGG11 based architecture achieved better accuracy. During the rebuttal period, we repeated CIFAR10 experiment for SENN with VGG19, which has very similar depth, more parameters than our network, but it performs worse than the reported experiment with VGG11.
>
>
>
> 7. The main reason why methods cited in L106-108 were not chosen as baselines and were separately mentioned in the related works is that they already assume availability of a ground truth dictionary of concepts for training. From the viewpoint of generating interpretations, learning a dictionary of concepts/attributes is the most crucial challenge for methods like FLINT or SENN, which is absent in these methods. Yes, we can explicitly add a mention about this in the main paper.

---

> > ### Comment · Reviewer_WJT2 · 2021-08-28
> > **Response to authors' rebuttal**
> >
> > I apologize for the late engagement. I thank the authors for the response. I am upgrading my score to 6. I am left with some thoughts/concerns though, and share them below for completeness:
> >
> > > Experimenting with more complex datasets warrants at least two modifications when applying FLINT: (1) Updating the decoder  architecture accordingly and, (2) Increasing the number of attributes  to accommodate larger number of classes and/or images.
> >
> > Thank you for the additional results, it's nice to see the results of CIFAR100. Considering this is an improvement over existing efforts in the space, it'd however have been really nice to see results on one complex dataset beyond the CIFAR datasets (e.g. ImageNet, TinyImageNet, CUB, etc). Please note that I am not doubting the correctness of the idea, it'd have been nicer to only be more convinced about its practical usefulness (since that's what seems to be motivating this work itself).
> >
> > > We discuss about this in supplementary Sec 2.1.1. We can create a separate subsection in supplements for this, if needed.
> >
> > I would recommend this, since this is important in a work where this can make a difference. It'd have ideally been nice to see what happens to the results when in a given setting, different layers are chosen. How much do the accuracy and fidelity (as well as other metrics) vary? The work seems a tad incomplete without such results.
> >
> > > We don't expect this choice (of which loss to introduce on which epoch) to have any major impact on a new dataset.
> >
> > It'd have been nice to have seen empirical evidence of what happens on when the loss is introduced at different epochs, even on a dataset of CIFAR10 scale.
> >
> > I'd be happy to discuss further if the authors have more to share, or if I missed something.

---

> > > ### Author Response · Authors · 2021-08-29
> > > **Response to comments of reviewer WJT2**
> > >
> > > We thank reviewer WJT2 for their positive update and useful suggestions. We add our response to the remaining concerns as following:
> > >
> > > Regarding the large-scale dataset experiment (comment 1), we will run experiments on TinyImageNet or CUB-200, however it will be difficult to obtain the complete results by September 2. In that case, we will add results on one of the datasets for the camera-ready version.
> > >
> > > Regarding the hidden layers and loss scheduling (comments 2, 3), we are running the experiments currently and will be happy to add empirical evidence for both comments.

---

> > > > ### Author Response · Authors · 2021-09-02
> > > > **Update: Experiments about hidden layers and loss scheduling**
> > > >
> > > > Posting this update as a separate comment to ensure everyone is notified.
> > > >
> > > > **(UPDATE, Sep 01)** We have completed experiments about (1) how choice of hidden layers can affect different metrics and (2) how introduction of $L_{of}, L_{cd}$ at different points of training can affect the metrics. We discuss their results below (they will be added in the supplements):
> > > >
> > > > (1) Experiment for hidden layers: With ResNet18 on QuickDraw. We make 3 different choices of single hidden layers (9th, 13th, 16th conv layers). For each choice we tabulate resulting metrics (accuracy, fidelity of interpreter, reconstruction loss, conciseness for threshold $1/\tau=0.2$). All other hyperparameters same.
> > > >
> > > > | Layer     	| acc (\%) $\uparrow$ 	| Fidelity of Interpreter (\%) $\uparrow$ 	| input fidelity $\downarrow$ 	| conciseness (threshold $1/\tau=0.2$)$\downarrow$ |
> > > > |-----------	|---------------------	|---------------------------------	|-----------------------------	|-----------------------------------------------------	|
> > > > | 9th conv  	| 85.2                	| 78.0                            	| 0.074                       	| 1.873                                               	|
> > > > | 13th conv 	| 85.6                	| 85.6                            	| **0.073**                   	| 1.905                                               	|
> > > > | 16th conv 	| 86.5                	| **96.0**                        	| 0.081                       	| **1.562**                                           	|
> > > >
> > > > **Key Observations**: (a) Compared to average BASE-$f$ accuracy of 85.3\%, accuracy of all models are comparable or slightly better. (b) The interpreter fidelity gets considerably better if the layer chosen is closer to the output. (c) The input fidelity/reconstruction loss does not behave as monotonously, but it is not surprising that layers close to the output result in worse input reconstruction. (d) Interpretations are expected to be more concise when chosen layer is very close to the output in the sense that conciseness is an indicator of abstraction level of the interpretation. Thus, a standard choice is to start with a layer close to the output. A small revision may be needed depending upon optimization of input fidelity loss.
> > > >
> > > > (2) Experiment for loss scheduling: With ResNet18 on CIFAR10. We introduce $L_{of}$ at different points of time in training (indicated by first column of table below). $L_{cd}$ is introduced 1 epoch later. The first row corresponds to current setting proposed in main paper. Total training time constitutes 25 epochs. All other hyperparameters remain same.
> > > >
> > > > | Time of introduction 	| acc (\%) $\uparrow$ 	| Fidelity of Interpreter (\%) $\uparrow$ 	| input fidelity $\downarrow$ 	| conciseness (threshold $1/\tau=0.2$)$\downarrow$ |
> > > > |----------------------	|---------------------	|---------------------------------	|-----------------------------	|-----------------------------------------------------	|
> > > > | Epoch 3 (current)    	| 84.6                	| 93.5                            	| 0.421                       	| 2.612                                               	|
> > > > | Epoch 4              	| 84.8                	| 93.4                            	| 0.427                       	| 2.501                                               	|
> > > > | Epoch 5              	| 84.3                	| 94.2                            	| 0.426                       	| 2.351                                               	|
> > > > | Epoch 6              	| 85.0                	| 93.1                            	| 0.426                       	| 2.376                                               	|
> > > > | Epoch 8              	| 84.5                	| 93.7                            	| 0.432                       	| 2.642                                               	|
> > > > | Epoch 10             	| 84.6                	| 93.9                            	| 0.422                       	| 1.944                                               	|
> > > > | Epoch 14             	| 84.2                	| 92.1                            	| 0.445                       	| 2.274                                               	|
> > > > | Epoch 21             	| 84.6                	| 91.2                            	| 0.450                       	| 3.710                                               	|
> > > > | Epoch 24             	| 84.4                	| 86.3                            	| 0.524                       	| 4.533                                               	|
> > > >
> > > > **Key Observations**: (a) As soon as the system gets reasonable time to train with all three losses (note that input fidelity loss is always present), small changes to introduction of losses has little to no impact on the metrics. (b) By contrast, when we introduce the losses extremely late (for example see the last two rows), the interpretability losses/metrics get noticeably worse.

---

### Decision · Program_Chairs · 2021-09-27

**Decision:**

Accept (Poster)

**Comment:**

All the reviewers agree that the paper has some nice and interesting ideas and it is indeed quite promising that the proposed method gives minimal accuracy loss while providing interpretability.

Having said that, a majority of them felt that a more thorough analysis of the method (e.g., ablation studies) and evaluation on at least one dataset on the scale of ImageNet is necessary to properly evaluate the utility of the proposed method. This is particularly so because of the large amount of existing work in this area and the limited novelty of the proposed method. I am sharing some of the key concerns and points that arose during the discussion period.

- As a problem on which there is earlier work (SENN, PrototypeDNN), restricting to small-scale datasets seemed limiting. The new results on CIFAR100 were nice to see, but results on one larger complex dataset (TinyImagenet, CUB, ImageNet) would have made it better to see the usefulness of the work (considering the method by itself has no new components, it's an aggregation of existing pieces).

- The paper seems to be missing analysis (ablation studies in the case of this work) where the results are shown with choice of different layers, or choice of different epochs where loss functions are introduced. These are important decisions in the framework, and how they impact the final result would have been nice to see (even if some of the choices did not yield strong results, and the method worked only for certain choices).

[Update] The authors' response dated 2nd September indeed satisfactorily addresses the two main issues of (i) evaluation on a larger dataset and (ii) analysis of the effect of different choices in the proposed framework. Since the reviewer/AC discussion had concluded by 30th August, this last authors' response was not noticed at the time of the original metareview. Given that all the reviewers agreed that the paper has interesting results, and that the two main criticisms have been satisfactorily addressed, I recommend acceptance of the paper.